# Heat is associated with short-term increases in household food insecurity in 150 countries and this is mediated by income

Carolin Kroeger ✉

Rising temperatures are expected to stall progress on food insecurity by reducing agricultural yields in the coming decades. But hot periods may also increase food insecurity within days when it gets too hot to work and earn an income, thus limiting households' capability to purchase food. Here I exploit variations in heat levels during a household survey spanning 150 countries in a quasi-natural experiment to show that particularly hot weeks are associated with higher chances of food insecurity among households (0.5767, 95% confidence interval 0.2958–0.8576, $t$ = 4.024, d.f. = 427,816, $P$ < 0.001). This association is mediated by reductions in income and health for households and the effects are stronger in countries with lower incomes and more agricultural or precarious forms of employment. The results highlight the importance of labour market disruptions for food insecurity and suggest integration of these concerns into heat action plans and food programmes.

Two billion people worldwide experience food insecurity[1]. Climate change is anticipated to stall progress towards food security, particularly in regions struggling with undernourishment already[2]. Existing evidence shows that rising temperatures may lead to food insecurity in the next months or years by reducing agricultural productivity and harvests[2–5]. But hotter temperatures may not only impact food availability in the long term but also immediately affect people's ability to access and afford food[6,7]. Particularly hot periods may lead to food insecurity within days when it gets too hot to work and households lose out on income, thus constraining their ability to afford food.

Hot periods create physical strain for workers and reduce their productivity or work hours which can create economic shocks for households[8–10]. For example, female brick workers in West Bengal, India, are paid by the number of bricks carried in a day and experience income losses of up to 50% when heat stress forces them to reduce their walking speed and carry fewer bricks[11]. Recent evidence from India documents the struggle of such workers to afford food when they are unable to work and earn an income[9]. Worldwide, ~470 billion potential work hours—equivalent to almost 1.5 weeks of work per person on Earth—were lost in 2021 to extreme heat with serious economic consequences that may affect food security[12].

This association between heat, income and food insecurity may differ across countries. People living in countries with more heat-exposed employment, such as construction or agriculture, may experience stronger effects of heat on food insecurity. For example, an estimated two-thirds of the potential work hours lost in 2021 accrued among agricultural workers[12]. Moreover, people living in countries with higher incomes may, on average, enjoy a higher standard of housing[13], access to cooling technologies[14] and higher affordability of food[15], which could protect people from heat exposure and help buffer the impact of income losses on food spending.

In this paper, I use a household survey across 150 countries to show that heat is associated with higher food insecurity within days of exposure and that this increase is mediated by reductions in income and health with stronger effects in countries with lower incomes and higher shares of agricultural or precarious employment. The analysis reveals a short-term income-based pathway between heat and food insecurity at the household level, suggesting a necessary addition to the current focus on long-term agricultural impacts in the academic debate and policy-making around heat and food insecurity.

Department of Social Policy and Intervention, University of Oxford, Oxford, UK. ✉e-mail: carolin.kroeger@spi.ox.ac.uk

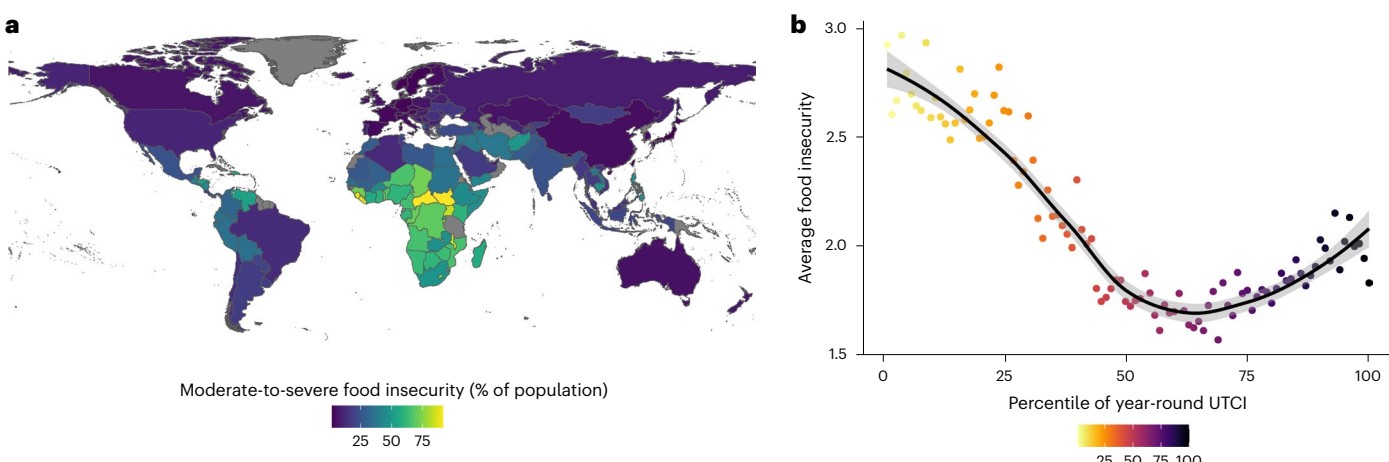

**Fig. 1 | Descriptive statistics for food insecurity and heat. a**, The average share of moderate-to-severe food insecurity in each country. **b**, The average level of food insecurity by year-round percentile of daily UTCI including a smoothed trend line with a 95% CI.

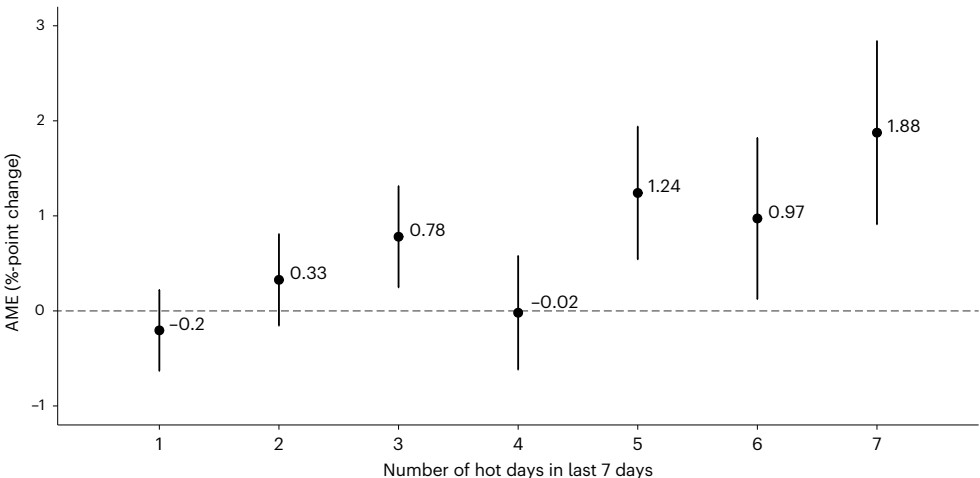

**Fig. 2 | Lagged effects of hot days on moderate-to-severe food insecurity.** AME of hot days in the last week on moderate-to-severe food insecurity; 95% CI are reported, *n* = 427,832.

## Results

The full sample in the analysis included 543,852 observations across 2,935 subregions and 150 countries (Fig. 1a). Moderate-to-severe food insecurity was reported by 25.75% of respondents with higher rates in countries with lower income. Across countries, the average level of food insecurity was lowest for the 50th to 75th percentile of year-round heat measured with the Universal Thermal Climate Index (UTCI, °C) but began increasing for higher percentiles (Fig. 1b). Almost 15% of all observations were made after a hot week which is over-representative of global experiences and reflects the more frequent surveying during the warmer months of the year.

The multilevel linear model regressing hot days against food insecurity shows that the number of hot days in the last week is associated with significantly higher levels of moderate-to-severe food insecurity (Supplementary Information 5.2). Figure 2 shows the effects of the number of hot days on moderate-to-severe food insecurity compared to not having experienced any hot days in the last week. The results suggest that the effects of heat accumulate over the course of the week and become stronger as the number of hot days in the week increases. For example, three hot days are associated with a 0.78%-point increase (95% confidence interval (CI) 0.2488–1.3119, *t* = 2.878, d.f. = 427,810,

*P* = 0.004) in chances of moderate-to-severe food insecurity compared to having experienced no hot days, whereas seven hot days are associated with a 1.88%-point increase (95% CI 0.9124–2.8383, *t* = 3.817, d.f. = 427,810, *P* < 0.001).

As a next step, I construct a binary indicator that classifies the last week as hot if at least three days were in the hottest 10% of the year in that subregion. This measure captures the accumulation of heat stress over several days displayed in Fig. 2 and simplifies the further analyses and interpretations. A hot week, on average, is associated with a 0.5406%-point increase (95% CI 0.2301–0.851, *t* = 3.413, d.f. = 543,847, *P* < 0.001) in chances of moderate-to-severe food insecurity across countries (Table 1, model 1). After controlling for area, gender, age, partner status, precipitation over the last week, average year-round UTCI, the number of hot days in the last 12 months, and fixed effects for the year, the effect increases to 0.7466%-points (95% CI 0.3979–1.0951, *P* < 0.001, d.f. = 427,816, *t* = 4.197) (Table 1, model 2). In the next step, covariate balancing propensity scores (CBPS) are used to minimize the covariate imbalance among the treatment and control group. Figure 3a shows the decrease in absolute mean differences for each covariate between the control and treatment group before and after covariate balancing. After CBPS-adjustment, a hot week is associated

**Table 1 | Models testing the relationship between a hot week and moderate-to-severe food insecurity**

| | Dependent variable: | | |
|---|---|---|---|
| | Moderate-to-severe food insecurity[a] | | |
| | (1) | (2) | (3) |
| Intercept | 0.2914 | 0.1215 | 0.1261 |
| | (0.2521, 0.3306) | (0.0827, 0.1603) | (0.0860, 0.1661) |
| | $P<0.0001$ | $P<0.0001$ | $P<0.0001$ |
| Hot week | 0.0054 | 0.0075 | 0.0058 |
| | (0.0023, 0.0085) | (0.0040, 0.0110) | (0.0030, 0.0086) |
| | $P=0.0007$ | $P<0.00013$ | $P=0.0001$ |
| Area: urban | | −0.0345 | −0.0331 |
| | | (−0.0374, −0.0317) | (−0.0360, −0.0302) |
| | | $P<0.0001$ | $P<0.0001$ |
| Age | | 0.0011 | 0.0012 |
| | | (0.0010, 0.0011) | (0.0011, 0.0012) |
| | | $P<0.0001$ | $P<0.0001$ |
| Partner: yes | | −0.0237 | −0.0236 |
| | | (−0.0261, −0.0213) | (−0.0260, −0.0212) |
| | | $P<0.0001$ | $P<0.0001$ |
| Gender: male | | −0.0167 | −0.0166 |
| | | (−0.0189, −0.0144) | (−0.0188, −0.0143) |
| | | $P<0.0001$ | $P<0.0001$ |
| Children: yes | | 0.0424 | 0.0377 |
| | | (0.0398, 0.0451) | (0.0351, 0.0404) |
| | | $P<0.0001$ | $P<0.0001$ |
| Hot days in the last 365 days | | 0.0061 | 0.0061 |
| | | (0.0050, 0.0071) | (0.0050, 0.0073) |
| | | $P<0.0001$ | $P<0.0001$ |
| Number of countries | 150 | 149 | 149 |
| Standard deviation of countries | 0.244 | 0.2031 | 0.2069 |
| Observations | 543,852 | 427,832 | 427,832 |
| Log likelihood | −233,823.8000 | −179,687.0000 | −242,523.4000 |
| Akaike information criterion | 467,657.5000 | 359,405.9000 | 485,078.8000 |
| Bayesian information criterion | 467,713.6000 | 359,581.4000 | 485,254.3000 |

[a]Moderate-to-severe food insecurity with (1) no controls, (2) controls and (3) controls and CBPS-adjustment for covariate imbalance. Two-sided $t$-tests and 95% CIs and $P$ values are reported under each coefficient. Year and precipitation variables are not shown in the table.

with a 0.5767%-point increase in chances of moderate-to-severe food insecurity (95% CI 0.2958–0.8576, $t = 4.024$, d.f. = 427,816, $P < 0.001$) (Table 1, model 3).

Figure 3b presents the unadjusted and CBPS-adjusted average marginal effects (AME) of a hot week on mild-to-severe, moderate-to-severe and severe food insecurity. The results show that the effect of a hot week becomes stronger for more severe forms of food insecurity. In addition, the CBPS weights consistently decrease the size but increase the precision of the estimated effect of a hot week.

Before proceeding to the mediation and moderation analysis, I conduct several robustness checks on the main model that delve into the assumptions and complexities behind the linear probability model for binary response. I provide a summary of the analyses here and report further details in Supplementary Information. As a first step, I test the assumptions underlying the quasi-natural experiment design. A key assumption is that households are randomly assigned into the treatment and control group so that, on average, the two groups are similar with respect to covariates. The CBPS method adjusts for

potential imbalances among treatment and control group, which often occur in quasi-natural experiments when treatment is not truly random. For example, Fig. 3b shows how CBPS-adjustment reduces the large difference in the average year-round heat among treated and non-treated households. I describe the CBPS procedure in more detail in Supplementary Information 3.1. However, the CBPS scores do not account for selection bias based on household income, which is a mediator and therefore excluded from the covariates in the main model. I find that households on lower incomes are more likely to be interviewed after a particularly hot week (0.0035, 95% CI 0.0016–0.0054, $t = 3.619$, d.f. = 460,146, $P < 0.001$). As a robustness check, I therefore include the share of lower income interviews on a day as a control variable in the main model and find that it does not substantially change the association between a hot week and food insecurity (0.0069, 95% CI 0.0034–0.0104, $t = 3.895$, d.f. = 427,815, $P < 0.001$) (Supplementary Information 3.2). I also test the model using more stringent precision requirements for the geographic match of the UTCI to the household, which may be less accurate for larger subregions. The positive and significant

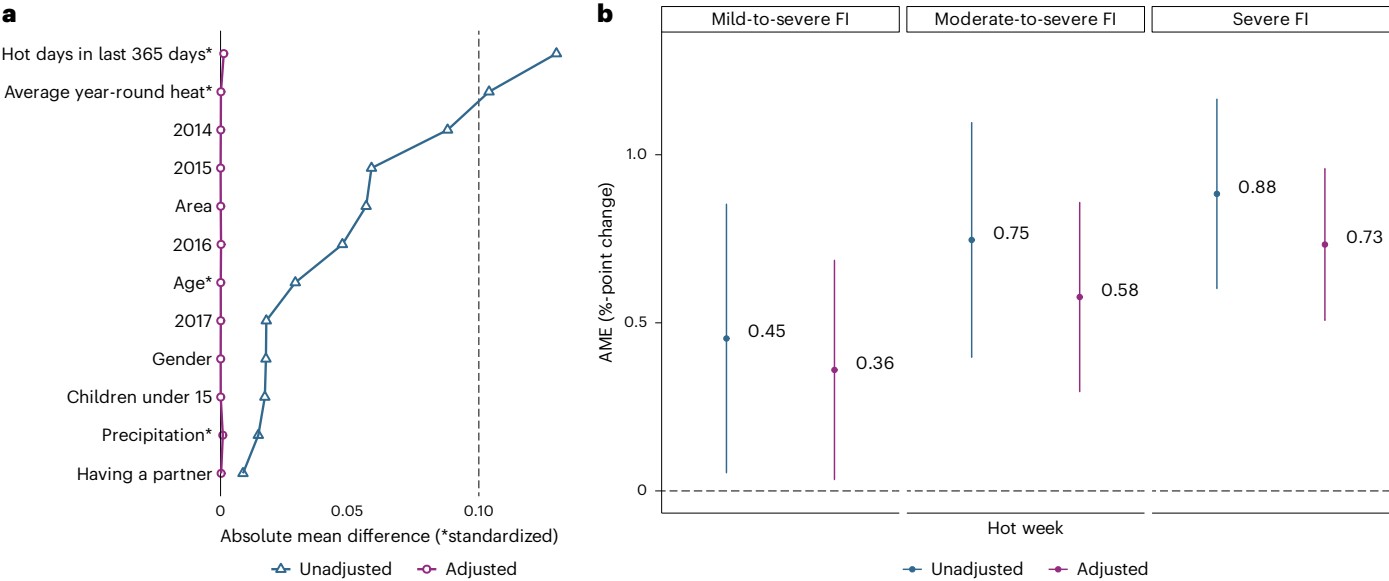

**Fig. 3 | Main results. a**, The reduction in absolute mean difference between the treatment and control group for each covariate before and after CBPS-adjustment. **b**, The adjusted and unadjusted AME of a hot week on varying degrees of food insecurity (FI) with 95% CIs; $n$ = 427,832.

association between a hot week and food insecurity hold even when controlling for the size and the variation of UTCI observations within subregions (0.0091, 95% CI 0.0056–0.0126, $t$ = 5.057, d.f. = 420,661, $P$ < 0.001). It also holds when using only subregions with a low variation (0.0114, 95% CI 0.0072–0.0156, $t$ = 5.293, d.f. = 281,053 $P$ < 0.001) (Supplementary Information 3.3). In Supplementary Information 4, I explore nonlinear model specifications and discuss the limitations of the data that restrict the model choices. I find that the linear probability model for binary response in the main manuscript outperforms more complex nonlinear models. For example, the Akaike information criterion for the linear probability model is 359,405 compared to 364,961 for the logit model (Supplementary Information 4.1). I also confirm the association between heat and food insecurity across continuous heat measures (Supplementary Information 4.2), absolute and relative heat measures (Supplementary Information 4.3), as well as different measures of food insecurity (Supplementary Information 4.3). In Supplementary Information 5, I investigate the lagged effects of heat on food insecurity in the long term and short term. I show that different ways of accounting for seasonality or previous heat exposure in the last 12 months do not substantially change the association between a hot week and food insecurity (Supplementary Information 5.1). I also present evidence that the effect of heat on food insecurity materializes within several days of heat exposure and that this effect dissipates for less recent exposures, suggesting that the observed changes in the 12-month measure of food insecurity are indeed reflective of changes in heat over the last few days leading up to the survey (Supplementary Information 5.2). Finally, I explore the sensitivity of the results to outliers and covariates and find that no country is an influential outlier in a DFBETA analysis. The maximum DFBETA is 0.0021 for Niger compared to a threshold of 0.0031 (Supplementary Information 6.1). I also find that the positive and significant association between a hot week and food insecurity holds using different sets of covariates (Supplementary Information 6.2). Overall, the analyses in Supplementary Information support the choice of a linear probability model for binary response as the main model and confirm that the results hold across different model specifications and measures of heat and food insecurity.

## Mediation analysis

Figure 4 presents the results from a mediation analysis that builds on the linear probability model for binary response. The results show

that income and health-related variables are significant mediators between a hot week and moderate-to-severe food insecurity. First, I test the association between a hot week and different mediators using a full set of demographic and weather-related control variables in a three-level mixed effects model with subregions nested within countries. The results show that individuals who have just experienced a hot week are significantly more likely to have health problems, experience difficulties living on their present income, report significantly lower income, give the local job market a worse rating and wanted more employment in the past seven days (Supplementary Information 2.1). Second, I sequentially test different mediators in two-level models for the country level and find that the average mediation effects are positive and significant for experiencing health problems (0.0596, 95% CI 0.0001–0.0009, d.f. = 422,027, $P$ < 0.001), difficulties getting by on income (0.0033, 95% CI 0.0022–0.0043, d.f. = 420,751, $P$ < 0.001), annual income (0.36, 95% CI 0.2939–0.4410, d.f. = 427,816, $P$ < 0.001), the local job market (0.0719, 95% CI 0.0452–0.1136, d.f. = 373,479, $P$ < 0.001) and positive but insignificant for wanting more employment (0.0247, 95% CI −0.0061, 0.0005, d.f. = 424,561, $P$ = 0.12). The income-related mediators account for most of the total effect of a hot week on food insecurity with 63.35% mediated by difficulties getting by on present income and 72.46% by annual income. The local job market rating mediates 16.26% and individual health problems account for 12.14%. The structural equation model framework used in this analysis was restricted to two-level models due to limitations in the packages available for mediation analysis which may lead to imprecision for the CIs and significance levels. For example, unemployment is not a significant mediator in a two-level model (0.0247, 95% CI −0.0061–0.0005, d.f. = 424,561, $P$ = 0.12), even though unemployment is significantly higher after a hot week in a three-level model (0.0041, 0.0010–0.0074, d.f. = 424,561, $t$ = 2.610, $P$ = 0.0091).

## Moderation analysis

To assess the heterogeneity in estimated effect size across countries, I interact different country-level moderators with the hot week indicator. The analysis shows that individuals may, on average, experience stronger, weaker or no effects of a hot week on food insecurity depending on their regions' economic and employment conditions. Figure 5 in the left panels shows the AME of a hot week on moderate-to-severe food insecurity at different levels of each moderator (Fig. 5a(i),b(i),c(i)).

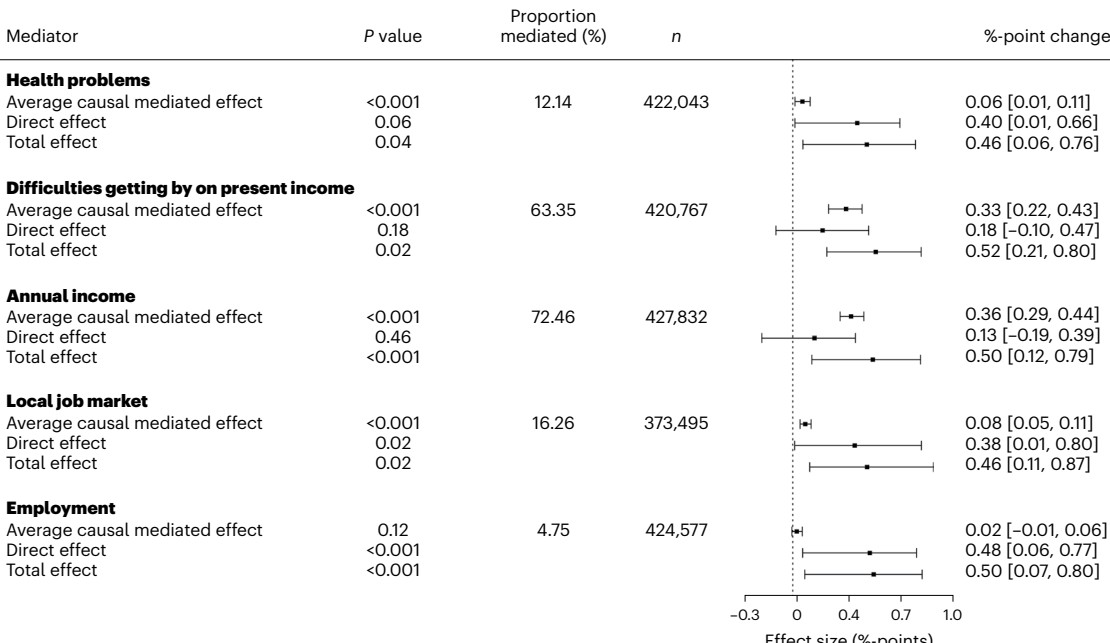

| Mediator | P value | Proportion mediated (%) | n | | %-point change |
|---|---|---|---|---|---|
| **Health problems** | | | | | |
| Average causal mediated effect | <0.001 | 12.14 | 422,043 | | 0.06 [0.01, 0.11] |
| Direct effect | 0.06 | | | | 0.40 [0.01, 0.66] |
| Total effect | 0.04 | | | | 0.46 [0.06, 0.76] |
| **Difficulties getting by on present income** | | | | | |
| Average causal mediated effect | <0.001 | 63.35 | 420,767 | | 0.33 [0.22, 0.43] |
| Direct effect | 0.18 | | | | 0.18 [−0.10, 0.47] |
| Total effect | 0.02 | | | | 0.52 [0.21, 0.80] |
| **Annual income** | | | | | |
| Average causal mediated effect | <0.001 | 72.46 | 427,832 | | 0.36 [0.29, 0.44] |
| Direct effect | 0.46 | | | | 0.13 [−0.19, 0.39] |
| Total effect | <0.001 | | | | 0.50 [0.12, 0.79] |
| **Local job market** | | | | | |
| Average causal mediated effect | <0.001 | 16.26 | 373,495 | | 0.08 [0.05, 0.11] |
| Direct effect | 0.02 | | | | 0.38 [0.01, 0.80] |
| Total effect | 0.02 | | | | 0.46 [0.11, 0.87] |
| **Employment** | | | | | |
| Average causal mediated effect | 0.12 | 4.75 | 424,577 | | 0.02 [−0.01, 0.06] |
| Direct effect | <0.001 | | | | 0.48 [0.06, 0.77] |
| Total effect | <0.001 | | | | 0.50 [0.07, 0.80] |

Effect size (%-points): −0.3 0 0.4 0.7 1.0

**Fig. 4 | Mediation analysis for the effect of a hot week on moderate-to-severe food insecurity.** Total, direct and average mediation effects and their 95% CIs for two-sided *t*-tests are reported for health problems (*n* = 422,043), difficulties getting by on present income (*n* = 420,767), the logarithm of annual household income (*n* = 427,832), the local job market (*n* = 373,495) and wanting more employment (*n* = 424,577). The number of simulations for each model is 100.

The left panels also report the density of observations at different moderator values in the data. The right panels in Fig. 5(a(ii),b(ii),c(ii)) show the countries expected to report positive and significant effects of a hot week on food insecurity given their level of the respective moderator. For example, Fig. 5a(i) shows that countries with a higher gross national income per capita experience, on average, significantly lower effects of a hot week on food insecurity. In fact, regions with a gross national income per capita higher than US$8,180 experience, on average, no positive and significant effects of a hot week on food insecurity. Figure 5a(ii) therefore shows regions with gross national incomes per capita lower than US$8,180, such as Egypt or Bolivia, which would be expected to experience statistically significant and positive effects of a hot week on food insecurity. I also find that regions with higher levels of agricultural or vulnerable employment as a share of total employment experience, on average, stronger effects of a hot week on food insecurity. Figure 5b(i) visualizes the significant and positive interaction between a hot week and agricultural employment (0.0005, 95% CI 0.0003–0.0007, *t* = 5.977, d.f. = 422,358, *P* < 0.001) on food insecurity. The model predicts that regions with less than 22% of agricultural employment, such as Russia or Mexico, are not expected to experience significant effects. Fig. 5b(i) also shows that a country with an agricultural employment share of 70% would, on average, experience a 2.79%-point increase in food insecurity after a hot week. However, the density plot of the observations reveals that there are few observations with such high levels of agricultural employment, suggesting that such strong effects may rarely be observed in practice. Finally, countries with higher levels of vulnerable employment, such as self-employed workers without employees or contributing family workers, experience stronger effects of a hot week on food insecurity (0.0003, 95% CI 0.0002–0.0004, *t* = 4.945, d.f. = 422,358, *P* < 0.001). The model predicts that countries with less than 32% of total employment classified as vulnerable, such as Slovakia or Botswana, are not expected to experience significant effects of a hot week. The heat maps indicate that a similar set of countries are affected by heat for each moderator. In addition to Fig. 5, I provide a table in Supplementary Information 1.3 with summary statistics for each country to help interpret whether a country has moderator values that, on average,

correspond to a significant and positive impact of a hot week on food insecurity in this model.

## Discussion

Hot weeks are associated with significantly higher chances of food insecurity. The CBPS-adjusted model predicts that if a country with the population of India were to experience a particularly hot week, an additional 8.07 million people would be likely to experience moderate-to-severe food insecurity. The analyses suggest that these effects are mediated by worse health, declining local job markets and tighter household budgets with stronger effects in regions with higher agricultural or vulnerable employment and lower incomes.

Academic debate and policy-making around heat and food insecurity has focused on medium- to long-term declines in agricultural yields[3–5]. Instead, here I show that heat can lead to higher food insecurity within days as heat-related physical strain and health problems limit peoples' ability to work and earn an income, thus limiting their ability to afford to buy food. This pathway emphasizes the affordability and accessibility of food and reintroduces a capability perspective into a debate that has largely focused on material food availability as a result of harvest shortages[7]. While this paper focused on the role of employment and household income, households may also experience rising expenditures since food spoils more easily in the heat, costs for cooling in the form of water or electricity may rise, and individuals may have to pay out-of-pocket for heat-related illnesses[16–18]. Cost-related effects that squeeze household budgets may be partially captured in the mediator variable that records difficulties getting by on income but further research should explore the cost-based pathways with more suitable cost data.

In line with the employment and income-based mediation, the moderator analysis showed that regions with larger shares of agricultural workers are more affected by hot weeks. These workers tend to be particularly heat-exposed and may experience stronger physical strain and heat-related productivity declines. This result also aligns with previous findings that agricultural workers account for about two-thirds of the heat-related work hour losses worldwide[12]. There could be residual confounding of the effect of agricultural employment by rural–urban

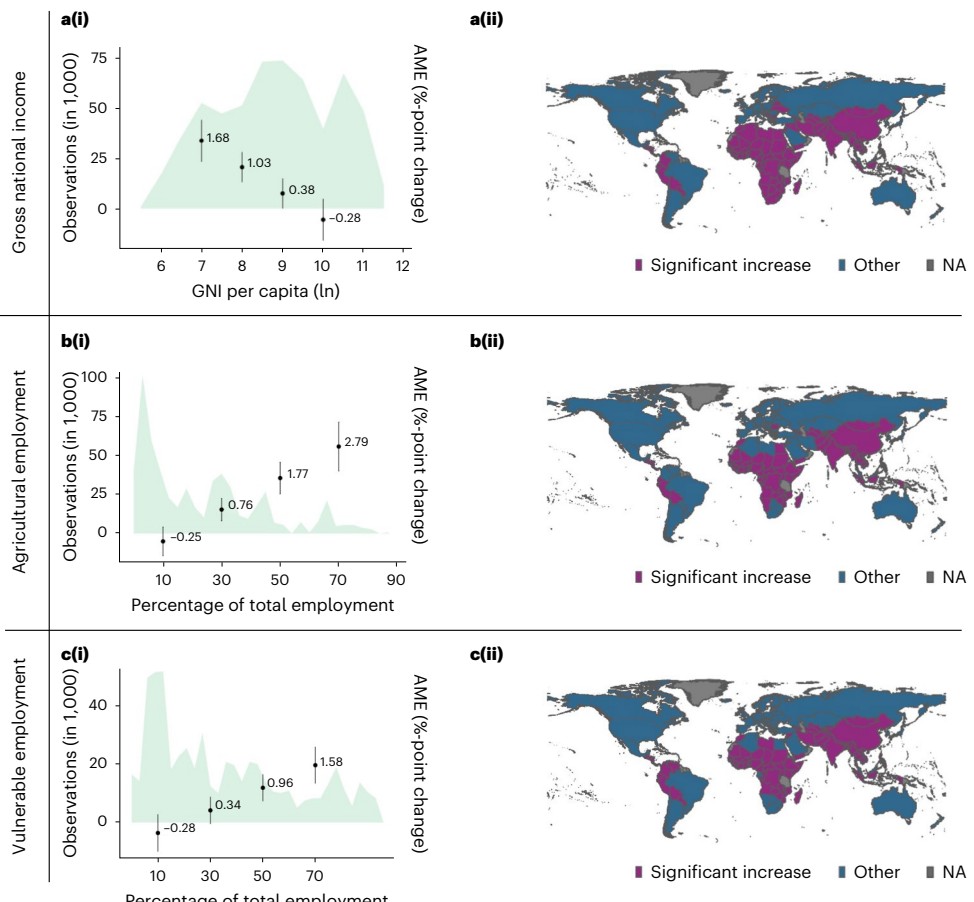

**Fig. 5 | Moderation analysis for the effect of a hot week on moderate-to-severe food insecurity. a–c**, The first column (i) shows the AME and their 95% CIs of a hot week on food insecurity at different levels of: gross national income (GNI) per capita (Atlas method, ln), n = 425,375 (**a**), agricultural employment as a share of total employment, n = 422,377 (**b**) and vulnerable employment as a share of total employment, n = 422,377 (**c**). The second column (ii) shows the countries that are predicted to experience, on average, a significant increase of a hot week on food insecurity based on the respective moderator value. Countries with no observations available are marked as NA (not available).

patterns in food access and supply chains. For example, people living in rural areas may have less access to different food sources, refrigerators and health care[5]. However, the model already controls for whether a respondent lived in an urban or rural area and the moderation effect of agricultural employment is robust to the inclusion of a country-level urbanization measure. Even though I use agricultural employment as an example for heat-exposed work, the effects of heat on food insecurity could also be stronger in other heat-exposed industries such as construction and may also more heavily affect workers in poorly ventilated factories, such as the textiles industry[19].

Employment conditions play a central role for the short run link between heat and food insecurity as underlined by the importance of vulnerable employment as a moderator. People living in countries with higher shares of vulnerable employment experience, on average, higher impacts of heat on food insecurity. Vulnerable employment is a measure by the International Labour Organization that captures contributing family workers and self-employed own-account workers without hired employees as a share of total employment. Vulnerable workers are most likely to fall into poverty, more likely to have informal work arrangements, least likely to have social protection or safety nets and often do not have sufficient savings to offset economic shocks[20]. These conditions may link their income more closely to their daily productivity and expose them to stronger income shocks[21,22]. In fact, a previous study found that hot temperatures are associated with absenteeism for salaried workers but not for workers on piece rate contracts, suggesting that salaried workers are able to stop working

because they do not experience a direct pay cut while piece rate workers may continue working to avoid an income loss[23].

Finally, regions with lower gross national incomes per capita experience, on average, higher effects of a hot week on food insecurity. Employment in richer countries may be less heat-exposed, require less manual labour and be less informal and therefore expose workers less to heat-related fluctuations in productivity and income. In addition, building standards and labour laws may help protect workers from intense heat exposure at the workplace and at home, where heat can also affect sleep and rest. Finally, people in richer countries are, on average, richer and may have higher savings and a stronger ability to buffer heat-related income shocks.

There are four main limitations in this analysis. The first limitation concerns geographical accuracy. The Gallup World Poll provides a geotag at the subregional level which in most countries corresponds to the first administrative layer, such as the state. While the median subregion is a little larger than half the size of Wales (12,434 km²), United Kingdom, there are also larger subregions, such as Texas, United States. For larger subregions, people at the border of two subregions may share more characteristics than people on opposite sides of the same subregion. This may violate the assumptions behind modelling intercepts at the subregional level nested within countries. In addition, the accuracy with which a UTCI averaged over the subregion reflects the local UTCI experienced by a household within larger subregions is likely to be lower. I tested these assumptions with different models and find that the results hold even when excluding subregions with

**Table 2 | FIES**

| | |
|---|---|
| 1 | You were worried you would not have enough food to eat? |
| 2 | You were unable to eat healthy and nutritious food? |
| 3 | You ate only a few kinds of food? |
| 4 | You had to skip a meal? |
| 5 | You ate less than you thought you should? |
| 6 | Your household ran out of food? |
| 7 | You were hungry but did not eat? |
| 8 | You went without eating for a whole day? |

**Table 3 | Visualization of the data structure**

| Country | Region | | | |
|---|---|---|---|---|
| United Kingdom | England | | Food insecurity index | Heat (°C) |
| | 1 May 2016 | Household 1 | 2 | 25 |
| | | Household 2 | 0 | 25 |
| | 4 May 2016 | Household 1 | 3 | 23 |
| | | Household 2 | 0 | 23 |
| | | Household 3 | 4 | 23 |
| | ... | | ... | ... |
| | Wales | | Food insecurity index | Heat (°C) |
| | 1 July 2016 | Household 1 | 3 | 20 |
| | | Household 2 | 2 | 20 |
| | 2 July 2016 | Household 1 | 0 | 18 |
| | | Household 2 | 5 | 18 |
| | | Household 3 | 0 | 18 |
| | ... | | ... | ... |
| India | Tamil Nadu | | Food insecurity index | Heat (°C) |
| | 22 August 2017 | Household 1 | 0 | 28 |
| | | Household 2 | 4 | 28 |
| | 23 August 2017 | Household 1 | 4 | 30 |
| | | Household 2 | 3 | 30 |
| | | Household 3 | 2 | 30 |
| | ... | | ... | ... |
| | Punjab | | Food insecurity index | Heat (°C) |
| | 14 September 2017 | Household 1 | 6 | 25 |
| | | Household 2 | 5 | 25 |
| | 16 September 2017 | Household 1 | 4 | 26 |
| | | Household 2 | 0 | 26 |
| | | Household 3 | 2 | 26 |
| | ... | | ... | ... |

high intrasubregion variation in UTCI and controlling for the variation of UTCI within subregions. However, further research should explore datasets with more granular geotags that also contain measures of food insecurity that are sensitive to short-term changes in household income and food consumption. Relatedly, further research using finer geographic scales should delve deeper into the precise heat thresholds at which different populations experience impacts of heat on health, income and food insecurity.

A second limitation is that the experiment is only quasi-natural. I used CBPS and checks on potential selection biases (Supplementary Information 3) to minimize the risk of confounding but in the absence of a truly randomized assignment, residual confounding cannot be ruled out entirely.

A third limitation is that the food insecurity experience scale (FIES) questions in the Gallup World Poll ask households to recount their experiences with food insecurity over the last 12 months but this study aims to test the short-term associations between a hot week and food insecurity (Table 2 for a list of questions in the FIES). I took several steps to ensure that the observed changes in food insecurity can be attributed to recent heat exposure in the last few days. First, I included the number of hot days in the previous 365 days before the survey in all models to isolate the impact of a recent hot week from more long-term heat exposures. I also control for different seasonal indicators and fit time-stratified models and find that the results between a hot week and food insecurity still hold (Supplementary Information 5.1). Finally, there is evidence in the literature that recency bias informs how people respond to questions about food insecurity[24,25]. In the mediation analysis I selected mediators such as whether households experience difficulties getting by on their present income which capture more recent changes in income. In addition, while the annual income variable references the entire year, this figure is calculated using the income reported over the last month. In combination, the analyses provide strong evidence that the variation in the 12 months food insecurity measure reflects short-term variations that can be explained by the recent hot week but further research with more recent measures of food insecurity at the household level is recommended. Another limitation concerns the modelling of the short-term effects of heat on food insecurity. I explored the effects of the number of hot days in the previous week (Fig. 2) and showed that more recent heat exposure plays a larger role than less recent heat exposure (Supplementary Information 5.2). However, further research should investigate the lagged effects in more detail using time-series data and methods such as distributed lag nonlinear models, which unfortunately are incompatible with the panel data in this paper (Table 3 visualizes the data structure).

A fourth limitation is that the models do not control for potential changes in food prices over time in response to heat. For example, the projected impacts of heat on harvests may increase anxieties about future food supply and increase prices instantaneously. Similarly, livestock may perish during particularly hot periods and increase prices for dairy and meat. While this is an important limitation, the effects of heat on food prices are likely to operate beyond the time frame of one week that is under study in this paper.

## Conclusion

Heat is associated with significant increases in food insecurity within days of exposure as heat-related impacts on income, health and local job markets reduce people's capability to earn an income and buy food. These effects are stronger in countries with lower incomes, higher agricultural employment and more vulnerable employment. The results underline the importance of labour market disruptions and socio-economic factors, such as precarious forms of labour, for food security and climate impact modelling. With the frequency, duration and intensity of extreme heat days rising across the world due to

climate change[26], researchers and policymakers across sectors should consider how the socio-economic links between heat, health, income and food insecurity can be integrated into research, heat action plans, food programmes and labour regulations.

## Methods

### Indicators

Data on food insecurity and sociodemographic information such as age, gender and income were collected by Gallup through face-to-face and phone interviews from 2014 to 2017. Different households were interviewed in each year during different times of the year. The interviews took place across 150 countries in subregions that typically corresponded to the first administrative layer, such as England or West Bengal. The sampling strategy was random and stratified by population size and geography. Samples were collected in proportion to state population within each country. The survey includes the FIES which captures whether respondents report different manifestations of not having sufficient resources to acquire food over the past 12 months. The FIES is a validated metric that is comparable across and within countries[27]. The scale includes eight 'yes' or 'no' questions that correspond to different levels of food insecurity (Table 2). Such questions include milder forms of food insecurity, such as 'were you worried you would not have enough food to eat?' and more severe forms such as 'whether your household ran out of food'. I process the eight questions into three binary measures of food insecurity. Severe food insecurity is defined as having at least seven 'yes' responses, moderate-to-severe food insecurity as having at least four 'yes' responses and mild-to-severe food insecurity as having at least one 'yes' response[28].

The weather dataset ERA5-HEAT is based on a reanalysis which models the Earth's climate at a spatial resolution of 0.25° × 0.25° longitude and latitude[29]. ERA5-HEAT includes the UTCI which measures the combined effect of air temperature, wind, humidity and radiation as the "air temperature (°C) of a reference environment that would elicit in the human body the same physiological response [...] as the actual environment"[30]. These features make the UTCI suitable for studies across microclimates and on physiological heat stress, for which the UTCI has been validated[30].

While the UTCI provides an accurate reflection of the current weather conditions, thermal comfort or how individuals perceive those conditions, is a highly subjective experience that depends on individual physiology, acclimatization and culture[31]. Accordingly, heat–health studies have used various approaches to model heat stress. In this design, I model heat with a relative approach that uses the percentile of the daily average UTCI in a year in a subregion. In other words, I consider the hottest days of the year in Wales, United Kingdom, and I consider the hottest days of the year in Tamil Nadu, India, although the absolute degree Celsius may differ substantially. This approach is in line with the World Meteorological Organization's definition of a heat wave as periods of unusually hot weather[32] and this approach is used in other cross-national studies of heat stress[33,34]. Moreover, this approach captures acclimatization and adaptation to local heat levels which helps individuals tolerate higher temperatures. For example, hospitalizations increase significantly at 27 °C in colder parts of the United States but only increase significantly above 40 °C in hotter parts of the United States where people are likely to be more acclimatized to hotter weather[35].

I therefore process the absolute UTCI observations from degree Celsius into a relative indicator of whether the previous seven days leading up to the date of the survey were unusually hot. I do so in four steps. First, I calculate the daily average UTCI in each subregion in the Gallup dataset. The median size of a subregion is 12,434 km² and the median standard deviation of UTCI observations within a subregion is 1.35 °C which supports the assumption that the area averages can be used as proxies for the local heat experienced by individual households within the subregion (Supplementary Information 1.5 and 3.3). Second,

I identify a hot day as one that falls into the hottest 10% of days in the year in that subregion. Third, I count the number of hot days in the seven days leading up to and including the date of the survey. Fourth, I define a hot week as one with at least three hot days. I use both the number of hot days in the last week and the binary hot week indicator in the statistical analysis. These indicators are simple ways to capture the accumulation of physiological heat stress and that individuals may be able to buffer one day of income losses but struggle with continued losses over several days[36]. Table 3 visualizes the resulting data structure after combining the Gallup and ERA5-HEAT data.

### Statistical models

I estimate different multilevel linear mixed models by restricted maximum likelihood. I model fixed effects for the covariates and I model random intercepts for subregions nested within countries. The key dependent variables are binary indicators of mild-to-severe, moderate-to-severe and severe food insecurity. The key independent variables are the number of hot days in the previous seven days and whether the last week was a hot week. Control variables include a set of demographic controls, specifically age, gender, partner status, whether there are children under 15 living in the household and whether the respondent lives in an urban or rural area. For categorical variables, the reference groups are being a woman, having no partner, no children, and living in a rural area. Weather-related covariates include precipitation in the previous week, year-round average UTCI in the subregion and the number of hot days a respondent experienced in the 365 days before the survey. I also include fixed effects for each individual year from 2014 to 2017. See Supplementary Information 1.1 for a full description of all variables. Significance tests in all analyses are two-sided t-tests.

These models exploit variations in heat in the days leading up to the survey at different times of the year and in different locations. These variations present a quasi-natural experiment where some households are randomly treated with a hot period and others are not. This interpretation rests on the assumption that the selection of households for the survey was random with regard to heat. To support this interpretation, I investigate the covariate imbalance between the treatment and control group and use CBPS to balance the covariates such that, on average, individuals with a similar propensity score have similar distributions of each covariate whether they received treatment or not[37,38]. I also check for selection bias into treatment and control group, explore the geographic precision of the heat observations for larger subregions and test nonlinearity. I report a summary of the results in the main manuscript and provide more detail in Supplementary Information 3.1–3.3 and 4.1.

I structure the main analysis in three parts. First, I fit different multilevel linear probability models that regress food insecurity on heat to test whether a global association exists. I test different degrees of severity for food insecurity and present models adjusted and unadjusted for covariate imbalance. I explore different time horizons for the effects of heat on food insecurity. Second, I conduct a mediation analysis using the R package mediation v.4.5 (refs. 37,38). A mediation analysis tests the variables that sit on the causal pathway between heat and food insecurity, such as income, employment and health. The mediation analysis uses structural equation modelling for linear models following ref. 39 and developed further in refs. 40,41. This approach to mediation fits a series of regression models to trace pathways between exposure, mediator and outcome. In this case, the models test the association between the hot week and each mediator, as well as the association between a hot week and food insecurity while controlling for the mediator. On the basis of these regression models, the mediation analysis disaggregates the direct effect of heat on food insecurity into the direct effect of heat and the indirect effect that is explained by changes in the mediating variable. The mediators tested include whether a respondent reports difficulties getting by on their present income, any health

problems that prevent them from doing things people their age would normally do, whether a respondent rates the local job market as good or bad and the logarithm of annual household income which is calculated on the basis of a monthly value reported by the household and likely to be a reflection of more recent household income levels. The mediation analysis used 100 simulations and complete observations for each mediator and calculated quasi-Bayesian CIs. The indirect effects are the average mediated effect across all simulations reported in this data. The analysis was restricted to modelling the data hierarchy at the country level due to limitations imposed by the software which may reduce the precision of standard errors and CIs. For the third and final step of the main analysis, I conduct a moderation analysis and interact country-level variables with the binary hot week indicator while controlling for gross national income per capita (ln). The results are presented as the AME at different levels of the interaction terms.

Finally, I conduct several robustness checks that I summarize in Results and report in more detail in Supplementary Information. These include testing experiment assumptions such as random assignment (Supplementary Information 3), exploring different ways of modelling heat including absolute versus relative measures, binary versus continuous variables and linear versus nonlinear models (Supplementary Information 4). I also investigate long-term and short-term effects of heat on food insecurity (Supplementary Information 5) and test sensitivity to outliers and covariate selection (Supplementary Information 6). Analyses were done in R v.4.3.0.

### Reporting summary

Further information on research design is available in the Nature Portfolio Reporting Summary linked to this article.

### Data availability

I combine representative microdata collected in the Gallup World Poll with weather data from the ERA5-HEAT data[42] by the European Centre for Medium-Range Weather Forecasts and the CPC Precipitation data by the National Oceanic and Atmospheric Administration (NOAA) Physical Sciences Laboratory in different multilevel models. The Gallup World Poll is available at charge from Gallup. ERA5-HEAT and CPC Precipitation data are publicly and freely available in the Climate Data Store and on the NOAA website. Data for the country-level moderators such as gross national income, vulnerable employment and agricultural employment are available from the World Development Indicators DataBank at the World Bank which compiles data from the International Labour Organization, the World Bank National Accounts data and the Organisation for Economic Co-operation and Development National Accounts data files. All data and code except for the Gallup World Poll are available from the author's GitHub page (ckroeger95) in the heat-fi-global repository.

### Code availability

Code is publicly available on GitHub at https://github.com/ckroeger95/heat-fi-global.git.

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

## Acknowledgements

I thank the workers who shared qualitative accounts of their experience with heat, work and food insecurity in different newspaper articles—in many ways, the income link is not my idea but their lived experience that I was able to trace quantitatively. I also thank A. Reeves, R. Khosla, C. Ruckteschler, B. Goodair, N. Muggleton, L. Sochas, A. Sauer, V. Govindarajan, T. Mehta and D. Matic for their insightful comments, help with code and words of encouragement. Finally, I thank the Rhodes Trust (Germany, 2020) for their support during my PhD. The funders had no role in study design, data collection and analysis, decision to publish or preparation of the manuscript.

## Author contributions

C.K. is the sole author of the study and conceptualized the study, reviewed literature, processed data, ran statistical models and responded to reviewer comments.

## Competing interests

The author declares no competing interests.

## Additional information

**Correspondence and requests for materials** should be addressed to Carolin Kroeger.

# Editorial Policy Checklist

This form is used to ensure compliance with Nature Portfolio editorial policies related to research ethics and reproducibility. For further information, please see our editorial policies site. All relevant questions on the form must be answered.

## Competing interests

Policy information about competing interests

In the interest of transparency and to help readers form their own judgements of potential bias, Nature Portfolio journals require authors to declare any competing financial and/or non-financial interest in relation to the work described in the submitted manuscript.

Competing interests declaration

☒ We declare that none of the authors have competing financial or non-financial interests as defined by Nature Portfolio.

☐ We declare that one or more of the authors have a competing interest as defined by Nature Portfolio.

## Authorship

Policy information about authorship

Prior to submission all listed authors must agree to all manuscript contents, the author list and its order and the author contribution statements. Any changes to the author list after submission must be approved by all authors.

☒ We have read the Nature Portfolio Authorship Policy and confirm that this manuscript complies.

Large Language Models (LLMs), such as ChatGPT, do not currently satisfy our authorship criteria. Notably an attribution of authorship carries with it accountability for the work, which cannot be effectively applied to LLMs. Use of an LLM should be properly documented in the Methods section (and if a Methods section is not available, in a suitable alternative part) of the manuscript.

☒ We confirm that the author list of this manuscript does not include any Large Language Models (LLMs).

Policy information about Authorship: inclusion & ethics in global research

All authors are encouraged to provide an "Inclusion & Ethics" statement where relevant.

☐ We have provided an "Inclusion & Ethics" statement.

## Data availability

Policy information about availability of data

Data availability statement

All manuscripts must include a data availability statement. This statement should provide the following information, where applicable:
   - Accession codes, unique identifiers, or web links for publicly available datasets
   - A description of any restrictions on data availability
   - For clinical datasets or third party data, please ensure that the statement adheres to our policy

☒ We have provided a full data availability statement in the manuscript.

Mandated accession codes (where applicable)
Confirm that all relevant data are deposited into a public repository and that accession codes are provided.

☒ All relevant accession codes are provided   ☐ Accession codes will be available before publication   ☐ No data with mandated deposition

## Code availability

Policy information about availability of computer code

**Code availability statement**

For all studies using custom code or mathematical algorithm that is deemed central to the conclusions, the manuscript must include a statement under the heading "Code availability" describing how readers can access the code, including any access restrictions. Code availability statements should be provided as a separate section after the data availability statement but before the References.

☒ We have provided a full code availability statement in the manuscript

## Data presentation

For all data presented in a plot, chart or other visual representation confirm that:

| n/a | Confirmed | |
|---|---|---|
| ☒ | ☐ | Individual data points are shown when possible, and always for $n \leq 10$ |
| ☐ | ☒ | The format shows data distribution clearly (e.g. dot plots, box-and-whisker plots) |
| ☐ | ☒ | Box-plot elements are defined (e.g. center line, median; box limits, upper and lower quartiles; whiskers, 1.5x interquartile range; points, outliers) |
| ☐ | ☒ | Clearly defined error bars are present and what they represent (SD, SE, CI) is noted |

## Image integrity

Policy information about image integrity

☒ We have read Nature Portfolio's image integrity policy and all images comply.

Unprocessed data must be provided upon request. Please double-check figure assembly to ensure that all panels are accurate (e.g. all labels are correct, no inadvertent duplications have occurred during preparation, etc.).

Where blots and gels are presented, please take particular care to ensure that lanes have not been spliced together, that loading controls are run on the same blot, and that unprocessed scans match the corresponding figures.

## Additional policy considerations

Some types of research require additional policy disclosures. Please indicate whether each of these apply to your study. If you are not certain, please read the appropriate section before selecting a response.

| Does not apply | Involved in the study |
|---|---|
| ☒ ☐ | Macromolecular structural data |
| ☒ ☐ | Unique biological materials |
| ☒ ☐ | Research animals and/or animal-derived materials that require ethical approval |
| ☒ ☐ | Human embryos, gametes and/or stem cells |
| ☒ ☐ | Human research participants |
| ☒ ☐ | Clinical data |
| ☒ ☐ | Archaeological, geological, and palaeontological materials |

I certify that all the above information is complete and correct.

Typed signature  Carolin Kroeger     Date  21 July 2023

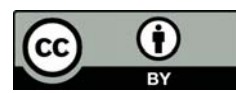

