## [Peer Review File · Nature Human Behaviour]

Peer Review Information

Journal: Nature Human Behaviour

Manuscript Title: Heat is associated with short-term increases in household food insecurity in 150 countries and this is mediated by income

Corresponding author name(s): Carolin Kroeger

Reviewer Comments & Decisions:

Decision Letter, initial version:

23rd January 2023

Dear Ms Kroeger,

Thank you once again for your manuscript, entitled "Do heat-related health and income losses increase food insecurity? A natural experiment across 148 countries, 2014-2017", and for your patience during the peer review process.

Your Article has now been evaluated by 3 referees. You will see from their comments copied below that, although they find your work of potential interest, they have raised quite substantial concerns. In light of these comments, we cannot accept the manuscript for publication, but would be interested in considering a revised version if you are willing and able to fully address reviewer and editorial concerns.

We hope you will find the referees' comments useful as you decide how to proceed. If you wish to submit a substantially revised manuscript, please bear in mind that we will be reluctant to approach the referees again in the absence of major revisions. We are committed to providing a fair and constructive peer-review process. Do not hesitate to contact us if there are specific requests from the reviewers that you believe are technically impossible or unlikely to yield a meaningful outcome.

To guide the scope of the revisions, the editors discuss the referee reports in detail within the team, including with the chief editor, with a view to (1) identifying key priorities that should be addressed in revision and (2) overruling referee requests that are deemed beyond the scope of the current study. We hope that you will find the prioritised set of referee points to be useful when revising your study. Please do not hesitate to get in touch if you would like to discuss these issues further.

1) Please address all reviewer concerns about the extent to which your analyses speak to short term effects rather than long term trends. We ask that you carry out additional analyses, as suggested by Reviewer 1, to adjust for long term effects.

2) We also ask that you address Reviewer 3's important concerns regarding the robustness of your relative approach in classifying hot weeks, and how meaningful or not this is in exploring mechanisms compared to an absolute measure.

3) Reviewer 1 raises concerns about your use of binary outcomes and we ask that you address these in full, following their suggestions for alternative analyses and presentation of results.

If you wish to submit a suitably revised manuscript we would hope to receive it within 4 months. I would be grateful if you could contact us as soon as possible if you foresee difficulties with meeting this target resubmission date.

- Include a "Response to the editors and reviewers" document detailing, point-by-point, how you addressed each editor and referee comment. If no action was taken to address a point, you must provide a compelling argument. When formatting this document, please respond to each reviewer comment individually, including the full text of the reviewer comment verbatim followed by your response to the individual point. This response will be used by the editors to evaluate your revision and sent back to the reviewers along with the revised manuscript.
- Highlight all changes made to your manuscript or provide us with a version that tracks changes.

[REDACTED]

Thank you for the opportunity to review your work. Please do not hesitate to contact me if you have any questions or would like to discuss the required revisions further.

Sincerely,

Charlotte Payne

Charlotte Payne, PhD
Senior Editor
Nature Human Behaviour

Reviewer expertise:

Reviewer #1: public health; heat and labour

Reviewer #2: health and climate change; economics; public health

Reviewer #3: public health; social policy; secondary analyses of large datasets

REVIEWER COMMENTS:

Reviewer #1:
Remarks to the Author:

The manuscript represents conceptual novelty in assessing for an association between food security and heat, with additional analysis to assess the impacts of income, agricultural and labour markets across countries. This has major socioeconomic and policy-related implications, particularly in the setting of global warming, and is highly original. The use of mediation analysis to assess associated factors in greater detail should also be commended.

The abstract is clear and accessible, although I would state the author's name in full under the Funding section of the abstract instead of as CK. Abstract, introduction and conclusions are appropriate. The use of references seems appropriate.

I feel there are two main flaws with the methodology. The first affects the validity of the results and is stated as a limitation:

Third, the food insecurity questions in the GWP cover the last 12 months, which may undermine our claim to a short-term association. However, recency bias strongly informs how people respond to questions about food insecurity and we therefore interpret the variables as more short-term 32,33

Although the links do show an association between short-term food security and 12-month assessment of food security, this is still an important limitation. This makes it difficult to distinguish the short-term effect of UTCI from its long-term effect, incorporating recency bias from both recent food security and recent heat. However, the model includes a variable for "average year-round UTCI in that subregion" variable. This would partially adjust for the long-term effect, although it only applies to that year instead of the last 12 months. To better adjust for the long-term effect over the last 12 months associated with the outcome, it would be more suitable to represent the year-round variable with different variable(s) represent UTCI over the last 12 months. Although this could be the average over the last 12 months, it would likely be more accurate to model the last 12 months in more detail. I would recommend a spline function for this, although periodic functions or a time-stratified model with month and year variables may also be suitable as per Bhaskaran et al. 2013 International Journal of Epidemiology <https://doi.org/10.1093/ije/dyt092>. The linked references in the manuscript give evidence that the effect varies non-linearly over the 30 days after SNAP receipt, which these

approaches may partially capture, or at least more effectively than a single yearly variable.

My second main concern with the methodological approach is the analysis of UTCI as a binary variable over a seven-day period, instead of as a continuous variable:

- This omits much of the useful information shown in a normally continuous variable (UTCI) by oversimplifying it
- This assumes each day in the 7-day period has an equal effect, which is likely not the case
- Binary variables have in the past been used to model heatwave days based on a predefined heatwave definition, although these are gradually being phased out in turn of definitions that include both continuous variable assessments of heatwaves and more detailed assessments for lag effects. This study does not seem to be particularly focused on heatwaves. Heatwave assessments are also generally limited to the hottest 4-6 months of the year in order to distinguish the effects of cold and heat. Although non-linear relationships can model both the heat and cold effects, binary and linear variables are likely to be confounded by the cold effect, which could increase or decrease food security compared to average UTCI.
- The current status quo method for assessing environmental exposure variables is with a distributed lag non-linear model. This models both the exposure-response and the lag-response relationships non-linearly. It offers a notably more flexible fit for both the immediate and delayed effects compared to a binary definition. of the results over the week.
 - o This would, however, affect how the results are presented, as they would not be able to be summarised by a single coefficient. This could be done by graphical presentation of results, presentation of the relative risk at a set percentile values (such as 90%) compared to a reference value (such as 50%), or as attributable risk (see Gasparrini and Leone 2014 for its calculation <https://dx.doi.org/10.1186/1471-2288-14-55>)
- Some papers highlighting the DLNM methodology with reproducible R code were published in 2015 by Gasparrini et al. 2015 ([https://doi.org/10.1016/s0140-6736\(14\)62114-0](https://doi.org/10.1016/s0140-6736(14)62114-0) and <http://dx.doi.org/10.1289/ehp.1409070>)

To a lesser degree, I also wonder why food insecurity was not assessed with further categories e.g. comparing mild, moderate and severe food insecurity as separate categories compared to no food insecurity. This would be more informative than assessing security as a binary variable with combined categories even if previous studies have taken this approach. Although this study does reference other studies that assessed food insecurity as a binary outcome, in particular Smith et al 2017, this study has a considerably large sample size that may provide sufficient power to distinguish between the four categories, particularly if a proportional odds assumption is valid (enabling a proportional odds logistic regression model).

The results and methodology need improvements in both presentation and structure, although I agree with the data interpretation. A subheading for the results section is missing. The Methods section should be after the Discussion for articles with Nature Human Behaviour. Some of the methodology description seems to be intertwined in the Results section instead of in the Methods section. Baron and Kenny's approach should be briefly explained, which is likely to be only be known to those familiar with mediation analysis.

Many of the results are not summarised or discussion, and are simply presented in tables or appendices (there is no mention of the results from appendices 3-6 in the main text, even briefly). The effects of columns (2) and (3) for hot days in Table 1 should be described in particular (I am also

unsure what these are). There is room for more descriptions in this manuscript (up to 5000 words for an Article, this manuscript is about 3000 words).

Up to eight figures and/or tables can be included in the main text. This has only three. Some of the tables/figures could be moved to the main text, in particular Appendix 3, as its results are described in about as similar detail as the other figures/tables.

Suggestions regarding the presentation of tables and figures:

- Generally
 - o The table could use some accompanying explanation. What do the numbers represent (they seem to be coefficients). What do the numbers in brackets represent (they seem to be standard error)? What do the columns represent (there are multiple tables with columns named as (1) to (4))?
- Table 1
 - o There are 3 columns summarising of the dependent variable in table 1, but I am not sure as to how they different mechanistically. Based on the text, it seems that the (1) outcome is the primary outcome. (2) and (3) may be the interaction effects with agricultural employment and informality, respectively, although I am unsure.
 - o It may be helpful to show more decimal places for the coefficients in the table, in particular hot week. For example, a hot week effect of 0.004 (0.002) could be non-statistically significant based on the rounding. This is only hinted by the ***, but providing even one more decimal place in this case would show it clearly. The level of detail is very clear, however, for its accompanying description in-text: "A hot week, on average, is associated with a 0.4217%-point increase [95%-CI: 0.1117 - 0.7317, p=0.0077, n=497,816]". This point can be applied to some other tables
 - o Log-likelihood and AIC BIC are presented to 3 decimal places, whereas they seem to be rounded to 1 or 2 decimal places.
- Figure 1. The title could be shortened with some of the text being moved as a legend description.
- Supplementary tables. The p descriptions are missing: *p***p***p<0.01. *p<0.05, **p<0.01, ***p<0.001
- Appendix 1
 - o Many of these variables seem to have very skewed distributions and would be better represented with median values and IQR instead of, or in addition to, mean and standard deviation; many of them.
 - o Some of the figure axes are small and difficult to read, such as Figures 1, 2 and 11. The most extreme example is the x-axis of Figure 1; said figure may need to be presented in landscape or reformatted?
 - o What do the red lines in Figure 1 represent?
 - o Figure 2. The y-axis would be better visualised in a different format than xe-xx i.e. 0 and 100,000 instead of 0e-00 and 1e-05.
 - o Table 4 should be separated by a page break from Table 3
 - o Figure 3. HH needs to be defined or either replaced by household
 - o Tables 5 and 6. I am not sure what (1)-(4) represent. Given that the row names and (1) are identical in both tables, perhaps they could be combined into one table presented in landscape mode, although it may be preferred to keep them separate
- P=0.0000 should be presented as P<0.0001

Some additional improvements include:

- Highlight which regions and/or countries are implicated by the conclusions e.g. those countries that have lower incomes, higher agricultural employment, and more informal labour markets

- Include descriptive summaries of the UTCI range per region as an appendix. This would both provide information about the exposure variable and highlight which countries are at greater risk from heat-related food insecurity.

Reviewer #2:

Remarks to the Author:

This study provides new evidence on how extreme heat affects food insecurity through reducing income and increasing health problem in the short term. The topic is new and interesting. However, I think several questions need to be further clarified before it could be considered for publication:

1. The FIES captures the food insecurity on year-scale, but the heat indicators (UTCI and hot week) are on daily scale. How does author match these two different scale data in the statistical models? As a non-expert in statistics, I doubt that simply collecting observation data which happened after a hot week can tell how many MORE people are likely to experience moderate-severe food insecurity through reductions in income and health. Besides, the extreme weather in the year t would reduce the food production and increase food insecurity in the year $t+1$, how do author exclude this impact when using FIES?

2. Several definitions in this study are unclear. (1) The author addresses a short-term impact in the study, however the definition of the short-term is not clear. The lag time for heat-related impacts on health and household income maybe different. Decrease of working capacity in workers with pieces salary would lead to a drop in income directly, but not necessarily to a health problem. (2) Please add explanation about "health variables", does it include all health outcomes?

3. Food storage and transportation would be difficult in extreme heat day, which could raise food price. The increasing of food price can also exacerbate food insecurity, especially for low-income group with high price elasticity of demand, which may lead to the same results in this study. Has the author considered or excluded this effect in the study?

4. The author finds that "The effects are stronger in countries with lower incomes, higher agriculture employment, and more informal markets". I think it would be more useful if the author could provide a figure to indicate which countries they are. Figuring out which country is more sensitive with the food insecurity problem associated with heat is important for policy makers.

5. The figure 2 (the bar on the far right) is incomplete, and a little blurry.

Reviewer #3:

Remarks to the Author:

This paper begins by laying out a strong rationale for investigating the short-term relationship between extreme heat, income shocks and food insecurity, supported by literature, and sets out to investigate this very important and timely research question. Robust statistical methods are used, as

are rich international datasets.

However, I am a bit concerned that this paper overstates its evidence of short-term impacts of hot days on food insecurity, income and health. Whilst significant findings are reported that are in line with expectations, especially the results from the interaction analyses, the outcome and main mediator variables here are not short-term measures. The food insecurity scale measures any experiences of food insecurity in the past 12 months; the income variable is a measure of annual income; and the measure of health is health problems, which could also reflect long-term health. Whilst significant findings between a short-term relative increase in temperatures and these outcomes are reported, it is not clear that these short-term increases are not reflecting longer term relationships. I would like to see the paper do more to convince the readers that short-term effects are truly being observed here, or more reflection that these findings do not rule out longer-term mechanisms also being at work and the need for more research/data that would allow short-term effects to be explored with greater precision.

Comments

1. Food insecurity in the FIES is recalled over a 12-month period. Data from the US show that when the same respondents answer the USDA Food Security Survey module with a 12-month and 30-day recall period, food insecurity levels for the 30-day recall period are about half that for a 12-month recall period. This suggests that a 12-month measure cannot be used to infer short-term food insecurity in the last 30 days, let alone in the last 7 days. Whilst this is briefly mentioned in the limitations section, I worry that this challenges the author's aim to examine the short-term relationship between heat days and food insecurity. Whilst this would be expected to bias results towards the null, given the lack of specificity of the outcome measure (i.e. potential for false positives- people classified as experiencing food insecurity in the short-term, where this could have been an experience from any point in the last 12 months), could it be possible that the relationship being observed here is reflecting a long-term one if the measure of a heat-day is related to longer-term measures of heat volatility? Has the author considered adjusting for total number of days with higher temperatures in their analysis?

2. Further, it is not clear that the method for classifying a hot week would actually capture climate conditions that match the mechanisms described in the introduction, namely, untenable heat conditions leading to inability to work, food spoilage, and heat stroke/other health outcomes. As a relative approach is used, depending on the country, it is possible that the days falling into the top 10% of hottest days are not actually that warm. This would be true in places with cool average temperatures and/or where conditions are stable all year round. I.e. the top 10% may not be that meaningfully different than the top 50%. This may be a misunderstanding of the method, but can more be said about why this relative approach was used, rather than an absolute approach identifying days where extreme heat was experienced? The interaction analysis later suggests that indeed, in countries with a lower average temperature, there is no effect of a hot week on food insecurity, which underscores that the main approach to classifying a hot week is not accurately capturing days with hot temperatures.

3. In the Methods section, it is not clear what level of geography is available in the Gallup World Poll data, which makes it difficult to follow the description of how heat shocks were quantified at the area-level. For example, it is suggested that England is a sub-region in the Gallup World Poll data but then that London is also a sub-region, which does not follow, since London is not mutually exclusive to

England. Please clarify what levels of geography are available in the Gallup data set. Importantly, please discuss the accuracy of the temperature data in relation to the sub-regions. Is it possible that there would be significant variation in temperatures within these sub-regions? I see this is mentioned in the limitations, but it would be useful to see this in discussion of the methods so that the reader better understands the heat day data available at the sub-regional level.

4. Please indicate data sources for variables used in interaction analyses – i.e. share in agricultural employment; informal labour market. The number of countries is significantly reduced in models including these interaction terms so it would be good to understand which countries had these data available and over what years.

5. Mediator variables: the premise of this paper is that short-term heat can lead to acute effects on income and health, leading to short-term food insecurity. Yet the mediators also do not capture short-term income or health effects all that well. The mediator variable that would seem to most specifically capture a short-term shock is wanting more employment in past 7 days but yet this seemed to mediate very little of the relationship observed? Does this not challenge the assertion that short-term effects are being observed here? Doesn't the mediation effect of annual income in relation to a measure of food insecurity in the last 12 months suggest that a longer-term relationship may be captured here?

6. Is there any data that would allow the author to consider the impact of household savings on this relationship? It would be expected that short-term heat shocks would have less impact on households that have access to liquid financial assets.

Minor comments

1. In abstract, suggest changing the phrase "knock-on" effects of climate shocks, as the observed effects would seem to be direct effects?
2. Final sentence of abstract: researchers don't tend to be involved in heat action and food security plans – sentence focus on policymakers, also perhaps NGOs?
3. Line 54: "But this focus on physical supply shortages..." - cut "physical" and insert "food" as in "food supply"
4. Line 74-75: Consider re-wording this sentence, as reads to suggest that construction or agriculture are less heat-exposed employment rather than examples of heat-exposed employment.
5. In description of Gallup World Poll data, please specify that new cross-sectional samples are drawn for each country each year. Please also indicate how many countries in each year participated and sample size in each year for majority of countries (i.e. 1000).
6. Please look at other descriptions of FIES and consider harmonising your description with how it is described in other papers. For example, the scale does not capture whether respondents experienced difficulties in accessing food, but rather manifestations of not having sufficient financial or other resources to acquire food – e.g. "...you were worried you would not have enough food to eat because of a lack of money or other resources?"; "...you had to skip a meal because there was not enough money or other resources to get food?". Consider including the questions in the appendix or in a box in the paper for readers that are unfamiliar with the scale.
7. Line 119-120 – reference is made to previous seven days. Specify that this is 7 days in relation to what? I think this is in relation interview date, but it is not stated that this information is collected/available in Gallup World Poll in section above.

Author Rebuttal to Initial comments

Reviewer 1

Reviewer Comment	Response
The manuscript represents conceptual novelty in assessing for an association between food security and heat, with additional analysis to assess the impacts of income, agricultural and labour markets across countries. This has major socioeconomic and policy-related implications, particularly in the setting of global warming, and is highly original. The use of mediation analysis to assess associated factors in greater detail should also be commended.	Thank you!
The abstract is clear and accessible, although I would state the author’s name in full under the Funding section of the abstract instead of as CK. Abstract, introduction and conclusions are appropriate. The use of references seems appropriate.	Done.
I feel there are two main flaws with the methodology. The first affects the validity of the results and is stated as a limitation: Third, the food insecurity questions in the GWP cover the last 12 months, which may undermine our claim to a short-term association. However, recency bias strongly informs how people respond to questions about food insecurity and we therefore interpret the variables as more short-term 32,33 Although the links do show an association between short-term food security and 12-month assessment of food security, this is still an important limitation. This makes it difficult to distinguish the short-term effect of UTCI from its long-term effect, incorporating recency bias from both recent food security and recent heat. However, the model includes a variable for	These are great points and thank you for referencing literature and including suggestions! I investigated your suggestion to use a spline function, but the application of the spline seems to be mostly for time series data. However, the data used here is a panel that interviewed households in different places at different times of the year. While these interviews often took place on consecutive days during roughly the same time of the year, the average number of days interviewed per year in a subregion is 5.5 days. We have researched applications of the spline to panel data but were unsuccessful in our search and tried to follow your suggestions in other ways: First, I follow your suggestion to control for the heat exposure over the last 12 months and

“average year-round UTCI in that subregion” variable. This would partially adjust for the long-term effect, although it only applies to that year instead of the last 12 months. To better adjust for the long-term effect over the last 12 months associated with the outcome, it would be more suitable to represent the year-round variable with different variable(s) represent UTCI over the last 12 months. Although this could be the average over the last 12 months, it would likely be more accurate to model the last 12 months in more detail. I would recommend a spline function for this, although periodic functions or a time-stratified model with month and year variables may also be suitable as per Bhaskaran et al. 2013 International Journal of Epidemiology <https://doi.org/10.1093/ije/dyt092>. The linked references in the manuscript give evidence that the effect varies non-linearly over the 30 days after SNAP receipt, which these approaches may partially capture, or at least more effectively than a single yearly variable.

calculate the number of hot days that an individual experienced in the 365 days leading up to their interview. I include this measure as a control variable in the main model and find that the positive and significant association of a hot week on food insecurity still holds. In fact, the effect size, standard error, and p-value remain largely unchanged. I also retain the average-year-round UTCI as a control variable since hotter countries tend to be poorer and more food insecure which may affect the relationship.

Second, in lieu of a spline function, I fit models with fixed effects for the year and the relative season and time-stratified models as outlined in Bhaskaran et al. 2013 for months nested within the year and the relative season nested within the year. The relative season identifies the hottest to coldest quartiles (or three months) of the year in a sub-region. I also follow your suggestion and fit a separate model using only the hottest six months of the year. The association between a hot week and moderate-to-severe food insecurity remains positive and significant across all models. The results are presented in Appendix 5.1. I interpret this as evidence that seasonal effects or trends, such as a higher level of food insecurity throughout the summer, are not driving the effects of a hot week on food insecurity.

I also want to clarify that some of the variables are calculated in a way that privileges recency. For example, the annual income is based on the monthly income that households report. Similarly, the mediator variable ‘feelings about household income’ specifically asks the household about their present income.

Finally, while these additional analyses provide strong evidence that the effect of a hot week can be isolated from more long-term trends or seasonal effects, I have included a short discussion of the limitations on page 16 and suggest avenues for future research.

My second main concern with the methodological approach is the analysis of UHCI as a binary variable over a seven-day period, instead of as a continuous variable:  • This omits much of the useful information shown in a normally continuous variable (UHCI) by oversimplifying it • This assumes each day in the 7-day period has an equal effect, which is likely not the case • Binary variables have in the past been used to model heatwave days based on a predefined heatwave definition, although these are gradually being phased out in turn of definitions that include both continuous variable assessments of heatwaves and more detailed assessments for lag effects. This study does not seem to be particularly focused on heatwaves. Heatwave assessments are also generally limited to the hottest 4-6 months of the year in order to distinguish the effects of cold and heat. Although non-linear relationships can model both the heat and cold effects, binary and linear variables are likely to be confounded by the cold effect, which could increase or decrease food security compared to average UHCI. • The current status quo method for assessing environmental exposure variables is with a distributed lag non-linear model. This models both the exposure-response and the lag-response relationships non-linearly. It offers a notably more flexible fit for both the immediate and delayed effects compared to a binary definition. of the results over the week.  o This would, however, affect how the results are presented, as they would not be able to be summarised by a single coefficient. This could be done by graphical presentation of results, presentation of the relative risk at a set percentile values (such as 90%) compared to a reference value (such as 50%), or as attributable risk (see Gasparrini and Leone 2014 for its calculation https://dx.doi.org/10.1186/1471-2288- 	Thank you for this comment and again for including alternative suggestions and literature. The DLNMs are a great reference and I already earmarked them for my next paper. I carefully read and replicated Gasparrini’s work, discussed it with several colleagues, and ran initial models on my data, but unfortunately came to the conclusion that the data structure in this study does not support DLNMs. This is because DLNMs are usually fitted on continuous time series (e.g., daily temperature and deaths in Chicago from 1990-2010). However, this data set includes several periods of consecutive or non-consecutive interview days (e.g., Daily food insecurity and temperature in Illinois in May 2014 and April 2015). I have included a visualisation of the data structure in the manuscript in Table 3. While I am not able to follow your specific recommendation for the DLNM, I fully appreciate the advantages of DLNMs compared to the initial analyses I presented and therefore conduct additional analyses to (1) model the lagged effect in more detail, (2) fit non-linear models, and (3) try continuous heat measures. First, I fit four different non-linear models: A logit model between moderate-severe food insecurity and the hot week indicator, and three Poisson models between the number of ‘yes’ responses to the FAO’s FIES scale (0-8) and the hot week indicator; the percentile of heat on the day of the survey; and the absolute heat on the day of the survey. In all cases, the effects of heat on food insecurity remain positive and significant (see Appendix 4.1). The non-linear probability model presented in the main manuscript has a comparable and even slightly lower AIC than the logit model, is computationally 55 times more efficient, and is easy to interpret by the range of
--	---

14-55)

•Some papers highlighting the DLNM methodology with reproducible R code were published in 2015 by Gasparrini et al. 2015 ([https://doi.org/10.1016/s0140-6736\(14\)62114-0](https://doi.org/10.1016/s0140-6736(14)62114-0)) and <http://dx.doi.org/10.1289/ehp.1409070>)

audiences that may be interested in this interdisciplinary work. I therefore choose to retain the linear model in the main manuscript but point readers towards the robustness checks in Appendix 4.1 on page 22 in the main manuscript.

Second, I investigate the lagged effects in more detail. I fit a regression between moderate-severe food insecurity and the number of hot days in the week before the survey date (see Appendix 5.2). The results show that the lagged effect increases, and confidence intervals become smaller with the number of hot days in the last week compared to having experienced no hot week. I also test the hot days of the last week in combination with hot days in the second-to-last week and find evidence that supports a focus on the last week (Appendix 5.2). This evidence supports the interpretation that several days of heat exposure over the last week are associated with higher levels of food insecurity. This interpretation is captured well in a binary heat variable which I therefore retain as the main model. However, I fully acknowledge that there is more nuance in the effect of heat than a binary variable allows for and therefore include the lagged effects of the last week in the main manuscript in Figure 3. I also extend the discussion section to discuss the lags in more detail page 16.

Third, I fit different models to explore continuous heat measures in Appendix 4.1 and 4.2. The percentile of UTCI on the day of the interview is associated with significantly higher levels of food insecurity in both linear and non-linear Poisson models. UTCI on the day of the interview is associated with significantly higher levels of food insecurity in non-linear Poisson models, but not in linear models. This is the result of the non-linear relationship between UTCI and food insecurity which becomes clear in the regression model that modelled UTCI on the day in 4°C buckets which finds positive and significant effects of UTCI for both colder and hotter bins.

In summary, I echo the concerns raised by the reviewer and hope to have addressed them by including more details on lagged short term effects in the main manuscript, demonstrating that non-linear versus linear modelling does not change the result that higher levels of heat are associated with higher levels of food insecurity, and presenting continuous heat measures as a robustness check in the appendix. Ultimately, I decided to retain a model that captures the intuition behind the theory and effectively communicates that exposure to short periods of hot temperatures will increase chances of food insecurity.

To a lesser degree, I also wonder why food insecurity was not assessed with further categories e.g., comparing mild, moderate and severe food insecurity as separate categories compared to no food insecurity. This would be more informative than assessing security as a binary variable with combined categories even if previous studies have taken this approach. Although this study does reference other studies that assessed food insecurity as a binary outcome, in particular Smith et al 2017, this study has a considerably large sample size that may provide sufficient power to distinguish between the four categories, particularly if a proportional odds assumption is valid (enabling a proportional odds logistic regression model).	Agreed. We have reported models using severe, moderate-to-severe, mild-to-severe, and a continuous measure of the food insecurity index (see Appendix 4.4). The effect sizes become larger and the confidence intervals smaller for more severe forms of food insecurity. We also report the average marginal effects of a hot week on severe, moderate-to-severe, and mild-to-severe forms of food insecurity in the main text in Figure 3 which shows that the effects become larger and confidence intervals smaller for more severe forms of food insecurity, although the overlapping of the confidence intervals suggests that these differences are likely not statistically significant.
The results and methodology need improvements in both presentation and structure, although I agree with the data interpretation. A subheading for the results section is missing. The Methods section should be after the Discussion for articles with Nature Human Behaviour. Some of the methodology description seems to be intertwined in the Results section instead of in the Methods section. Baron and Kenny's approach should be briefly explained, which is likely to only be known to those familiar with mediation analysis.	Done.
Many of the results are not summarised or discussion and are simply presented in tables or appendices (there is no mention of the results from appendices 3-6 in the main text, even briefly). The effects of columns (2) and (3) for hot days in Table 1 should be described in particular (I am also unsure what these are). There is room for more descriptions in this manuscript (up to 5000 words for an Article, this manuscript is about 3000 words).	Very well taken. I have moved some results from the Appendix into the main text and added more details, rationales, and examples throughout the manuscript.
Up to eight figures and/or tables can be included in the main text. This has only three. Some of the tables/figures could be moved to the main text, in particular Appendix 3, as its results are described	Agreed. I added and revamped the figures and tables to help visualise the descriptive statistics and data set as well as the results.

in about as similar detail as the other figures/tables.	
Suggestions regarding the presentation of tables and figures:  •Generally  o The table could use some accompanying explanation. What do the numbers represent (they seem to be coefficients). What do the numbers in brackets represent (they seem to be standard error)? What do the columns represent (there are multiple tables with columns named as (1) to (4))? •Table 1  o There are 3 columns summarising of the dependent variable in table 1, but I am not sure as to how they differ mechanistically. Based on the text, it seems that the (1) outcome is the primary outcome. (2) and (3) may be the interaction effects with agricultural employment and informality, respectively, although I am unsure. o It may be helpful to show more decimal places for the coefficients in the table, in particular hot week. For example, a hot week effect of 0.004 (0.002) could be non-statistically significant based on the rounding. This is only hinted by the ***, but providing even one more decimal place in this case would show it clearly. The level of detail is very clear, however, for its accompanying description in-text: “A hot week, on average, is associated with a 0.4217%-point increase [95%-CI: 0.1117 - 0.7317, p=0.0077, n=497,816]”. This point can be applied to some other tables o Log-likelihood and AIC BIC are presented to 3 decimal places, whereas they seem to be rounded to 1 or 2 decimal places. •Figure 1. The title could be shortened with some of the text being moved as a legend description. •Supplementary tables. The p descriptions are missing: *p***p***p<0.01. *p<0.05, **p<0.01, ***p<0.001 •Appendix 1  o Many of these variables seem to have very skewed distributions and would be better 	Thank you so much for this attention to detail. I revised the presentation of tables and figures with your suggestions as well.

represented with median values and IQR instead of, or in addition to, mean and standard deviation; many of them.  o Some of the figure axes are small and difficult to read, such as Figures 1, 2 and 11. The most extreme example is the x-axis of Figure 1; said figure may need to be presented in landscape or reformatted? o What do the red lines in Figure 1 represent? o Figure 2. The y-axis would be better visualised in a different format than xe-xx i.e. 0 and 100,000 instead of 0e-00 and 1e-05. o Table 4 should be separated by a page break from Table 3 o Figure 3. HH needs to be defined or either replaced by household o Tables 5 and 6. I am not sure what (1)-(4) represent. Given that the row names and (1) are identical in both tables, perhaps they could be combined into one table presented in landscape mode, although it may be preferred to keep them separate •P=0.0000 should be presented as P<0.0001 	
Some additional improvements include:  •Highlight which regions and/or countries are implicated by the conclusions e.g., those countries that have lower incomes, higher agricultural employment, and more informal labour markets •Include descriptive summaries of the UTCI range per region as an appendix. This would both provide information about the exposure variable and highlight which countries are at greater risk from heat-related food insecurity. 	Thanks, these suggestions are great to improve the policy relevance of this paper. As suggested, I include a table in Appendix 1.3 that shows country profiles with the UTCI range per region, the average absolute UTCI value lying behind the 90th percentile threshold in each country, as well as important moderators such as GNI, agricultural, and informal employment. Additionally, I added ‘heat maps’ in Figure 5 in the main manuscript that identify the countries that fall into the vulnerable range given their moderator variable. For example, the average marginal effects indicate that countries with agricultural employment higher than 22% of total employment are vulnerable to heat. I therefore indicate those countries with agricultural

	employment higher than 22% of total employment in the heat map.
--	---

Reviewer 2

Reviewer Comment	Response
1. The FIES captures the food insecurity on year-scale, but the heat indicators (UTCI and hot week) are on daily scale. How does author match these two different scale data in the statistical models? As a non-expert in statistics, I doubt that simply collecting observation data which happened after a hot week can tell how many MORE people are likely to experience moderate-severe food insecurity through reductions in income and health. Besides, the extreme weather in the year t would reduce the food production and increase food insecurity in the year t+1, how do author exclude this impact when using FIES?	Thanks for this point! You've mentioned that you consider yourself a non-expert in statistics, so I will lead with more intuitive explanations of the analyses we conducted. You are exactly right that there is a mismatch between the timeframe (=12 months) on which the FIES is captured and the timeframe for the heat indicator (= 1 week). The issue this poses is that trends or seasonal fluctuations, as well as past exposure to heat over the last 12 months may bias the association between the last week of heat and food insecurity. As you mention, for example, some years may have been more food insecure than others for reasons unrelated to the temperature in the last week, and this may bias results for that year. I therefore fit different models using different covariates and approaches to make sure that the effect of a hot week is not biased: First, I calculate the heat exposure over the last 12 months as the number of hot days that an individual experienced in the 365 days before the survey. Including this variable isolates the effect of a hot week from more long-term heat exposure. In this model, the effect of a hot week remains positive and significant and largely unchanged in terms of size and standard error.

Second, I include fixed effects for the year and the season, which I define as the hottest, second-hottest, second coldest, and coldest three months. The fixed effects for the year capture any year-to-year variation in food insecurity. For example, the larger size and significance of the fixed effect of the year 2017 (see Table 1 in the main manuscript) suggests that food insecurity was higher that year compared to the reference year 2014, which aligns with the FAO's estimation of global food insecurity from 2014-2017⁸. This procedure also accounts for the issue around harvest reductions in the last year that you mention in your second point. The seasonal variable accounts for seasonal variations within the year. If, for example, food insecurity was always highest in the summer months, then the hot week indicator may inadvertently capture this seasonal increase. Including a seasonal control variable is a way to counter this and try to isolate the effect of a hot week from seasonal fluctuations. I also fit different time-stratified models following the suggestions of another reviewer⁷. I first fit a model time stratified for months within the year (e.g., November 2014, December 2014). This model is rank deficient, meaning that the number of predictors in the model is too large given the number of observations. I therefore fit a model using the relative season nested within the year (e.g., second hottest season 2014, hottest season 2014...). In both models the effect of a hot week on food insecurity remains positive and significant with a small decrease in size compared to the main model reported in the manuscript (see Appendix 5.1).

I also clarify in the manuscript that some of the mediator variables are calculated in a way that privileges recency. For example, the annual income is based on the monthly income that households report during the survey. In addition, another mediator variable that captures whether

	households experience difficulties getting by on their income specifically refers to present income. Overall, these additional analyses isolated the effect of a hot week from more long-term influences on food insecurity and support the interpretation of a short-term effect. I have also added a section in the discussion page 16 to discuss the short-term effects and the length of the lags modelled and suggest avenues for future research.
2. Several definitions in this study are unclear. (1) The author addresses a short-term impact in the study, however the definition of the short-term is not clear. The lag time for heat-related impacts on health and household income maybe different. Decrease of working capacity in workers with pieces salary would lead to a drop in income directly, but not necessarily to a health problem. (2) Please add explanation about “health variables”, does it include all health outcomes?	I have clarified that the lag refers to the last week throughout the manuscript and included a discussion section on the lagged effects (page 16). I also included more detailed definitions of the mediating variables. In response to your specific question: The health variables is a yes or no response to ‘Do you have any health problems that prevent you from doing any of the things people your age normally can do?’. To clarify the causal pathway, our argument is that heat leads to physical distress which forces people to reduce their work intensity in the heat. While it would be interesting to investigate in a future article, I am currently not trying to show that the decrease in work capacity leads to health problems.
3. Food storage and transportation would be difficult in extreme heat day, which could raise food price. The increasing of food price can also exacerbate food insecurity, especially for low-income group with high price elasticity of demand, which may lead to the same results in this study. Has the author considered or excluded this effect in the study?	Great point! Initially when I designed this work, I thought that heat could increase food insecurity in the short term by (A) decreasing income because of worker productivity, and (B) increasing costs because of food spoilage and increase in prices, health expenditure for clinic visits, and higher costs for water or energy.

	I had initially included a brief outline of the cost-based mechanism in the introduction section, but we have now decided to remove it and mention these alternative pathways in the discussion section on page 13. There are three main reasons. First, most of the effect of a hot week on food insecurity could be explained through income reductions in the mediation analysis. Second, I do not have any data on costs at the household level in this data set. Third, the cost mechanisms operate on more complex time frames, transmission mechanisms, and local context than this data and study can do justice. For example, there have been reports of the price of chicken in West India rising by 35% within a month after an extreme heat wave, but how, when, and for whom this price increase occurs and how it interacts with other heat-induced price shocks is unclear. I am currently working on a follow up piece using a different data set to investigate this further, but it is unfortunately not possible within this paper.
4. The author finds that “The effects are stronger in countries with lower incomes, higher agriculture employment, and more informal markets”. I think it would be more useful if the author could provide a figure to indicate which countries they are. Figuring out which country is more sensitive with the food insecurity problem associated with heat is important for policy makers.	Fully agreed. I added a ‘heat map’ for each of the indicators to show which countries I would expect to be affected given their levels of agricultural employment, GNI per capita, or vulnerable employment in Figure 5 in the manuscript. I also included a table with country profiles with summary statistics for variables relevant for the interpretation such as UTCI, agricultural employment or vulnerable employment. See appendix 1.3.
5. The figure 2 (the bar on the far right) is incomplete, and a little blurry.	Well taken. Figures have been revised to ensure readability.

Reviewer 3

Reviewer Comment	Response
I am a bit concerned that this paper overstates its evidence of short-term impacts of hot days on food insecurity, income, and health. Whilst significant findings are reported that are in line with expectations, especially the results from the interaction analyses, the outcome and main mediator variables here are not short-term measures. The food insecurity scale measures any experiences of food insecurity in the past 12 months; the income variable is a measure of annual income; and the measure of health is health problems, which could also reflect long-term health. Whilst significant findings between a short-term relative increase in temperatures and these outcomes are reported, it is not clear that these short-term increases are not reflecting longer term relationships. I would like to see the paper do more to convince the readers that short-term effects are truly being observed here, or more reflection that these findings do not rule out longer-term mechanisms also being at work and the need for more research/data that would allow short-term effects to be explored with greater precision.  1. Food insecurity in the FIES is recalled over a 12-month period. Data from the US show that when the same respondents answer the USDA Food Security Survey module with a 12-month and 30-day recall period, food insecurity levels for the 30-day recall period are about half that for a 12-month recall period. This suggests that a 12-month measure cannot be used to infer short-term food insecurity in the last 30 days, let alone in the last 7 days. 	Thank you for this point, it really helped me improve the paper. I conducted several analyses in response to this comment that isolate the impact of the hot week from seasonal impacts or more long-term exposures to heat over the last 12 months. First, I followed your suggestion and calculated a count of the ‘heat exposure’ over the last 12 months. I defined this as the number of hot days (90th percentile in the year in the subregion) that an individual has experienced in the 365 days leading up to the survey. I include this variable as a control in the main model and find that the results hold: The effect of a hot week on food insecurity is still positive and significant and the effect size and confidence interval do not change substantially. I retain this control variable throughout all models presented in the analysis. Second, I fit two different models that include fixed effects for the months, and fixed effects for the relative seasons, which I calculated as the hottest, second-hottest, second coldest, and coldest months of the year. The results hold across both models and the effect of a hot week on food insecurity remains positive, significant, and comparable in terms of effect size and standard errors (Appendix 5.1). I also fit a separate model using only the hottest six months of the year and find that the results hold, but that the standard errors have increased which is likely linked to the smaller sample size (172,537 compared to >422k). Finally, I fit time-stratified models with strata for the month/year as outlined in Bhaskaran et al. 2013. Since the model is rank-deficient, I also fit a time-stratified model for the relative

Whilst this is briefly mentioned in the limitations section, I worry that this challenges the author’s aim to examine the short-term relationship between heat days and food insecurity. Whilst this would be expected to bias results towards the null, given the lack of specificity of the outcome measure (i.e. potential for false positives- people classified as experiencing food insecurity in the short-term, where this could have been an experience from any point in the last 12 months), could it be possible that the relationship being observed here is reflecting a long-term one if the measure of a heat-day is related to longer-term measures of heat volatility? Has the author considered adjusting for total number of days with higher temperatures in their analysis?	season/year. For both models, the results hold (Appendix 5.1). I clarify in the manuscript on page 22 that some of the variables are calculated in a way that privileges recency. For example, the annual income is based on the monthly income that households report in the survey. In addition, another mediator variable that captures difficulties with household income references the feelings of the household about their present income. Overall, these analyses isolated the effect of a hot week from seasonality, trends, or more long-term exposure to heat over the last 12 months and find that the effect of a hot week on food insecurity has remained positive, significant, and largely unchanged in terms of size or significant for all models. This provides strong evidence for an interpretation of the effect of a hot week as short term. In addition to referring readers to these robustness checks in the Appendix, I also added a section in the discussion around the limitations of the 12 months measure and the short-term effects on page 16.
2. Further, it is not clear that the method for classifying a hot week would capture climate conditions that match the mechanisms described in the introduction, namely, untenable heat conditions leading to inability to work, food spoilage, and heat stroke/other health outcomes. As a relative approach is used, depending on the country, it is possible that the days falling into the top 10% of hottest days are not actually that warm. This would be true in places with cool average temperatures and/or where conditions are stable all year round. I.e., the top 10% may not be that meaningfully different than the top 50%. This may be a misunderstanding of the method, but can more be said about why this relative approach was used, rather than an absolute approach identifying days where extreme heat was experienced? The interaction analysis later suggests that indeed, in	This is a great point and unfortunately still a largely unsolved puzzle in the heat-health literature. In terms of theory, there is no perfect way to model heat stress because thermal comfort is a highly subjective experience that depends on individual physiology, acclimatization, and culture¹. I argue that a relative approach is preferable over an absolute for three main reasons: First, a relative approach allows for acclimatization to heat which means that people living in hotter climates for longer are physiologically able to tolerate higher heat levels because their bodies have adapted to

countries with a lower average temperature, there is no effect of a hot week on food insecurity, which underscores that the main approach to classifying a hot week is not accurately capturing days with hot temperatures.

higher heat stress. For example, hospitalizations increase significantly at 27°C in colder parts of the US but 40°C in hotter parts of the US.² An absolute threshold across countries does not allow for modelling this regional adaptation. A relative modelling of heat stress is in line with the World Meteorological Organization's definition of a heat wave as periods of statistically unusual hot weather³, and is used in other cross-national studies of heat stress^{4,5}.

Second, the relative thresholds represent absolute heat levels that could reasonably be considered strenuous for heat-exposed or physical labour. For more than 50% of the observations, the absolute UTCI value underlying the 90th percentile of year-round daily UTCI is higher than 26.6°C (see Appendix 4.3). This average daily UTCI can reasonably be expected to be strenuous for heat-exposed or manual workers, especially given that the daily maximum UTCI is likely to be even higher than the average of 26.6°C.

Third, a relative measure is likely to yield a more conservative estimate because some weeks will be considered hot that were rather cool. For example, hot weeks in Iceland were on average based on days crossing 12°C. These misclassifications may also explain why countries with lower year-round temperatures no longer experience significant effects in the moderator analysis. As you rightly point out, in these countries the highest percentiles are simply not hot. However, these misclassifications are likely to bias the estimate towards zero. It is therefore an approach to modelling heat that errs on the side of caution. In contrast, an

absolute threshold-based approach is likely to misclassify countries in a different way. Countries with hotter year-round temperatures will qualify most observations as hot weeks, whereas countries with lower year-round temperatures will likely qualify none, even though heat stress is a documented issue in colder regions as well ⁶. Since a relative threshold identifies reasonably ‘hot’ observations for the majority of the observations and the misclassifications only lead to a more conservative estimate, I decide that theoretical considerations speak for the relative measure. I decided to remove the results around the climate moderation from the main text because I agree that they seem counter intuitive at first and shift focus away from the socio-economic factors this analysis focuses on.

While theory supports a relative threshold, I empirically test both absolute and relative heat modelling and different thresholds in linear and non-linear models and see that the results hold across a range of models.

For relative heat measures, heat has a positive and significant effect on food insecurity when using the 85th, 90th, and 95th percentile to define a hot week (Appendix 4.3). Heat is also positive and significantly associated with food insecurity when using percentile on the day of the interview in a poisson and a linear regression model (Appendix 4.1 and 4.2).

For absolute heat measures, I test different absolute thresholds (26°C, 28°C, and 30°C) to define a hot week and find positive but insignificant effects of heat on food insecurity. These appear to be linked to high multicollinearity between the hot week indicator and the year-round UTCI which

	affects the size and precision of the estimate (Appendix 4.3). For example, the Pearson correlation between year-round UTCI and a hot week indicator using a 26°C threshold is 0.67. Indeed, when excluding the year-round UTCI as a covariate, all absolute heat thresholds have positive and significant effects on food insecurity (Appendix 4.3). I also test UTCI on the day of the survey and find that the effects in the linear model are negative and insignificant (Appendix 4.2). This is likely linked to the non-linear effects of absolute UTCI on food insecurity. Binning UTCI on the day of the survey into 4°C-bins shows that both hotter and colder bins are associated with significant increases in food insecurity compared to days with 14-18°C. Using the continuous and absolute measure of UTCI on the day in a more suitable non-linear model shows that the association with a hot week is positive and significant (Appendix 4.1) Overall, theoretical reasons support the choice of a relative measure of heat, and the empirical analysis showed that the results for heat and food insecurity hold across different absolute models. I included the rationale for choosing a relative measure in the methods section of the main manuscript on page 18-19 and in Appendix 4.3.
3. In the Methods section, it is not clear what level of geography is available in the Gallup World Poll data, which makes it difficult to follow the description of how heat shocks were quantified at the area-level. For example, it is suggested that England is a sub-region in the Gallup World Poll data but then that London is also a sub-region, which does not follow, since London is not mutually exclusive to England. Please clarify what levels of geography are available in the Gallup data set. Importantly, please discuss the accuracy of the temperature data in relation to the sub-regions. Is it possible that there would be	This is unfortunately an inherited inconsistency from the Gallup survey which usually followed an administrative subregion one below the country-level, but also included a few context-specific sub-regions such as London, even though it is not mutually exclusive to England. In response to your comment, I have clarified this inconsistency in the appendix (3.3) and provided more detail on the size of sub-regions in the methods section on page 15.

significant variation in temperatures within these sub-regions? I see this is mentioned in the limitations, but it would be useful to see this in discussion of the methods so that the reader better understands the heat day data available at the sub-regional level.	In response to your comment about the accuracy of the geographic match, I have also in Appendix 3.3 included a boxplot for the area of a subregion and show that the median size of a sub-region is 12,434 km², which is a little larger than half the size of Wales, United Kingdom. I also plot the longitude and latitude points within each sub-region of the United Kingdom to give an indication of how dense the longitude and latitude observations are relative to the sub-regions. Additionally, I included a histogram in Appendix 1.5 Panel B) with the variation of UTCI across the longitude and latitude points within a sub-region. The average standard deviation of UTCI within an area is 1.81°C with a median of 1.35°C. Finally, I test whether the precision of the geographic match influences the results in two separate regression models in Appendix 3.3. Model (1) uses only areas with a standard deviation of UTCI within the area that is less than 2°C and still reports positive and significant effects on food insecurity with a comparable standard error and a substantially larger effect size. Model (2) controls for the standard deviation of UTCI within an area and the area size in km² (ln) and also reports a positive and significant association between a hot week and moderate-severe food insecurity. Combined, these models are evidence that the results hold when placing more stringent requirements around the geographic precision on the model (Model 1) and isolating the effect of a hot week from the variation and size of the area (Model 2).
4. Please indicate data sources for variables used in interaction analyses – i.e. share in agricultural employment; informal labour market. The number of countries is significantly reduced in models including these interaction terms so it would be good to understand which countries had these data available and over what years.	Well taken. I added the data sources in the methods section on page 17 and in the Appendix 1.1. I also replaced the informal economy indicator with an indicator for the share of vulnerable labour in the economy which is available for a much higher number of countries and more tailored to the theory around labour and employment that I laid out.

5. Mediator variables: the premise of this paper is that short-term heat can lead to acute effects on income and health, leading to short-term food insecurity. Yet the mediators also do not capture short-term income or health effects all that well. The mediator variable that would seem to most specifically capture a short-term shock is wanting more employment in past 7 days but yet this seemed to mediate very little of the relationship observed? Does this not challenge the assertion that short-term effects are being observed here? Doesn't the mediation effect of annual income in relation to a measure of food insecurity in the last 12 months suggest that a longer-term relationship may be captured here?	I included the number of hot days an individual experienced in the last 365 leading up to the survey as an additional covariate in the mediation analysis as well. This should help isolate the short-term effect of a hot week from those of past heat experience. I also clarified in the manuscript that some of the variables are calculated in a way that indicates recency. For example, the annual income is based on the monthly income that households report. In addition, the mediator capturing difficulties getting by on income references the feelings of the household about their present income. Finally, the mediation analysis is limited in that the current R packages only support hierarchical data with two levels (= countries in our case), while we would ideally model the sub-regions nested within countries. This leads to slight inaccuracies around significance levels. For example, a hot week is associated with significantly higher levels of un- or underemployment in mixed effect linear models with regions nested within countries. However, that significance gets lost when using it within the mediation package and its restrictions on hierarchies which affects clustering of standard errors. We have included this as a limitation in the discussion and suggested avenues for further research.
6. Is there any data that would allow the author to consider the impact of household savings on this relationship? It would be expected that short-term heat shocks would have less impact on households that have access to liquid financial assets.	Interesting point! I went back to the original data set to see if I could retrieve data on this but unfortunately it only had data from one single country (<15k from China) on the savings-related questions, so I was not able to produce a meaningful analysis within the cross-national design in this paper. Out of interest, we

	experimented with models using that data for China and found that people who were more satisfied with their savings experienced weaker effects of a hot week on food insecurity but none of the effects were significant.
Minor comments 1. In abstract, suggest changing the phrase “knock-on” effects of climate shocks, as the observed effects would seem to be direct effects?	I removed this from the abstract in favor of a more specific statement of the contribution of this work: ‘The results highlight the importance of labour market disruptions for climate impact modelling around food security and suggest integration of these concerns into heat action plans and food programmes.’
2. Final sentence of abstract: researchers don’t tend to be involved in heat action and food security plans – sentence focus on policymakers, also perhaps NGOs?	I’ve changed the language in the abstract to this: ‘The results highlight the importance of labour market disruptions for climate impact modelling around food security and suggest integration of these concerns into heat action plans and food programmes.’ I have also updated the conclusion to reflect this: ‘Researchers and policymakers across sectors should consider how the socio-economic links between heat, health, income, and food insecurity can be integrated into research as well as heat action plans, food policy programs, and labour policies.’
3. Line 54: “But this focus on physical supply shortages...” - cut “physical” and insert “food” as in “food supply”	Done.
4. Line 74-75: Consider re-wording this sentence, as reads to suggest that construction or agriculture	Done. Good catch.

are less heat-exposed employment rather than examples of heat-exposed employment.	
5. In description of Gallup World Poll data, please specify that new cross-sectional samples are drawn for each country each year. Please also indicate how many countries in each year participated and sample size in each year for majority of countries (i.e. 1000).	I included a better description and visualisation of the data in the manuscript (see Table 3). I also included a summary table with country profiles in the Appendix (1.3) that shows the number of observations for important variables such as agricultural employment or temperatures including the sample size. Appendix 1.1 also provides an overview about the numbers of observation per year.
6. Please look at other descriptions of FIES and consider harmonising your description with how it is described in other papers. For example, the scale does not capture whether respondents experienced difficulties in accessing food, but rather manifestations of not having sufficient financial or other resources to acquire food – e.g. “...you were worried you would not have enough food to eat because of a lack of money or other resources?; “...you had to skip a meal because there was not enough money or other resources to get food?”. Consider including the questions in the appendix or in a box in the paper for readers that are unfamiliar with the scale.	Done. We included a table (Table 2) in the manuscript with the FIES questions.
7. Line 119-120 – reference is made to previous seven days. Specify that this is 7 days in relation to what? I think this is in relation interview date, but it is not stated that this information is collected/available in Gallup World Poll in section above.	Done.

Decision Letter, first revision:

13th June 2023

Dear Dr. Kroeger,

Thank you for your patience as we've prepared the guidelines for final submission of your Nature Human Behaviour manuscript, "Heat increases food insecurity unequally in the world by reducing income." (NATHUMBEHAV-22112970A). Please carefully follow the step-by-step instructions provided in the attached file, and add a response in each row of the table to indicate the changes that you have made. Please also check and comment on any additional marked-up edits we have proposed within the text. Ensuring that each point is addressed will help to ensure that your revised manuscript can be swiftly handed over to our production team.

We would hope to receive your revised paper, with all of the requested files and forms within two-three weeks. Please get in contact with us if you anticipate delays.

Nature Human Behaviour offers a Transparent Peer Review option for new original research manuscripts submitted after December 1st, 2019. As part of this initiative, we encourage our authors to support increased transparency into the peer review process by agreeing to have the reviewer comments, author rebuttal letters, and editorial decision letters published as a Supplementary item. When you submit your final files please clearly state in your cover letter whether or not you would like to participate in this initiative. Please note that failure to state your preference will result in delays in accepting your manuscript for publication.

In recognition of the time and expertise our reviewers provide to Nature Human Behaviour's editorial process, we would like to formally acknowledge their contribution to the external peer review of your manuscript entitled "Heat increases food insecurity unequally in the world by reducing income.". For those reviewers who give their assent, we will be publishing their names alongside the published article.

Cover suggestions

As you prepare your final files we encourage you to consider whether you have any images or illustrations that may be appropriate for use on the cover of Nature Human Behaviour.

ORCID

Non-corresponding authors do not have to link their ORCID but are encouraged to do so. Please note that it will not be possible to add/modify ORCIDs at proof. Thus, please let your co-authors know that if they wish to have their ORCID added to the paper they must follow the procedure described in the following link prior to acceptance:

Nature Human Behaviour has now transitioned to a unified Rights Collection system which will allow our Author Services team to quickly and easily collect the rights and permissions required to publish your work. Approximately 10 days after your paper is formally accepted, you will receive an email in providing you with a link to complete the grant of rights. If your paper is eligible for Open Access, our Author Services team will also be in touch regarding any additional information that may be required to arrange payment for your article. Please note that you will not receive your proofs until the publishing agreement has been received through our system.

Please note that *Nature Human Behaviour* is a Transformative Journal (TJ). Authors may publish their research with us through the traditional subscription access route or make their paper immediately open access through payment of an article-processing charge (APC). Authors will not be required to make a final decision about access to their article until it has been accepted. Find out more about Transformative Journals

[REDACTED]

Best regards,
Alex McKay
Editorial Assistant
Nature Human Behaviour

On behalf of

Charlotte Payne

Charlotte Payne, PhD
Senior Editor
Nature Human Behaviour

Reviewer #1:
Remarks to the Author:

Thank you for your comments and addressing my feedback or attempts with explanations where this was not possible. I feel with the additional sensitivity analyses conducted, there is enough evidence to support the association found with hot days and that this is likely to remain true when considering factors not directly included in the main model such as non-linearity and lag.

Reviewer #2:
Remarks to the Author:

Thanks for addressing my comments and nice revisions. In particular, I appreciate that the authors explicitly mentioned the limitations on cost-based mechanism in the discussion section. However, I am still concerned that neglecting the increased cost of food caused by heat could increase the degrees of attribution of income-based mechanism to food insecurity. In a short term, sticky wage may make it difficult for companies to reduce wages even if there is a declining in labor productivity. In contrary, increasing food price due to increased cost is generally more flexible. Is it possible to examine the effect chains of income by controlling for changes in food price? I'll leave it to the editor to decide whether further clarification on these points is worthwhile.

Reviewer #3:
Remarks to the Author:

The author has done an excellent job of responding to all reviewers' comments, providing robust sensitivity analyses and strong justifications of the approaches taken. Well done.

Author Rebuttal, first revision:

Reviewer #1 (Remarks to the Author):

Thank you for your comments and addressing my feedback or attempts with explanations where this was not possible. I feel with the additional sensitivity analyses conducted, there is enough evidence to support the association found with hot days and that this is likely to remain true when considering factors not directly included in the main model such as non-linearity and lag.

Response to reviewer #1

Thank you!

Reviewer #2 (Remarks to the Author):

Thanks for addressing my comments and nice revisions. In particular, I appreciate that the authors explicitly mentioned the limitations on cost-based mechanism in the discussion section. However, I am still concerned that neglecting the increased cost of food caused by heat could increase the degrees of attribution of income-based mechanism to food insecurity. In a short term, sticky wage may make it difficult for companies to reduce wages even if there is a declining in labor productivity. In contrary, increasing food price due to increased cost is generally more flexible. Is it possible to examine the effect chains of income by controlling for changes in food price? I'll leave it to the editor to decide whether further clarification on these points is worthwhile.

Response to reviewer #2

Thanks for raising this point! I think you are exactly right that heat increases food prices, but this would likely play out on a longer time frame (=weeks to months) than the income mechanism I describe (=days). In addition, you rightly point out that sticky wages make it harder to reduce wages. This argument is captured in the moderation analysis which shows that these effects are stronger in countries with more vulnerable or more informal employment where wages are more easily adjusted or dependent on worker's productivity. I explore this more in the discussion. There is also an issue around data availability for food prices/affordability on a global scale (all 150 countries) and local level (subregions within countries). I've raised the cost-based mechanism as a limitation in the discussion and suggested further research on this. I unfortunately believe it is not possible to address this concern within the limitations of the data and the declared scope of this paper. I am currently working on a follow up piece on cost-based impacts using daily household banking data and hope to explore the cost-based mechanism in a separate paper focused on that.

Reviewer #3 (Remarks to the Author):

The author has done an excellent job of responding to all reviewers' comments, providing robust sensitivity analyses and strong justifications of the approaches taken. Well done.

Response to reviewer #3

Thank you!

Final Decision Letter:

Dear Ms Kroeger,

We are pleased to inform you that your Article "Heat is associated with short-term increases in household food insecurity in 150 countries and this is mediated by income.", has now been accepted for publication in *Nature Human Behaviour*.

Please note that *Nature Human Behaviour* is a Transformative Journal (TJ). Authors whose manuscript was submitted on or after January 1st, 2021, may publish their research with us through the traditional subscription access route or make their paper immediately open access through payment of an article-processing charge (APC). Authors will not be required to make a final decision about access to

their article until it has been accepted. IMPORTANT NOTE: Articles submitted before January 1st, 2021, are not eligible for Open Access publication. Find out more about Transformative Journals

With best regards,

Charlotte Payne

Charlotte Payne, PhD
Senior Editor
Nature Human Behaviour